# The Inflammatory Response after Moderate Contusion Spinal Cord Injury: A Time Study

**DOI:** 10.3390/biology11060939

**Published:** 2022-06-20

**Authors:** Minna Christiansen Lund, Ditte Gry Ellman, Maiken Nissen, Pernille Sveistrup Nielsen, Pernille Vinther Nielsen, Carina Jørgensen, Ditte Caroline Andersen, Han Gao, Roberta Brambilla, Matilda Degn, Bettina Hjelm Clausen, Kate Lykke Lambertsen

**Affiliations:** 1Department of Neurobiology Research, Institute of Molecular Medicine, University of Southern Denmark, 5000 Odense C, Denmark; minnacl@hotmail.com (M.C.L.); dellman@health.sdu.dk (D.G.E.); maiken92@live.dk (M.N.); pernillesveistrup@hotmail.com (P.S.N.); pvnielsen@health.sdu.dk (P.V.N.); carinajoergensen@hotmail.com (C.J.); rbrambilla@med.miami.edu (R.B.); bclausen@health.sdu.dk (B.H.C.); 2Department of Clinical Research, University of Southern Denmark, 5000 Odense C, Denmark; dandersen@health.sdu.dk; 3Andersen Group, Department of Clinical Biochemistry, Odense University Hospital, 5000 Odense C, Denmark; 4Danish Center for Regenerative Medicine, Odense University Hospital, 5000 Odense C, Denmark; 5Department of Spine Surgery, Third Affiliated Hospital of Sun Yat-sen University, Guangzhou 510630, China; gaoh35@mail.sysu.edu.cn; 6Guangdong Provincial Center for Engineering and Technology Research of Minimally Invasive Spine Surgery, Guangzhou 510630, China; 7The Miami Project to Cure Paralysis, Miller School of Medicine, University of Miami, Miami, FL 33136, USA; 8Brain Research Inter-Disciplinary Guided Excellence (BRIDGE), Department of Clinical Research, University of Southern Denmark, 5000 Odense C, Denmark; 9Department of Pediatrics and Adolescent Medicine, Rigshospitalet, 2100 Copenhagen, Denmark; matildadegn@gmail.com; 10Department of Neurology, Odense University Hospital, 5000 Odense C, Denmark

**Keywords:** neuroinflammation, cytokines, tumor necrosis factor, immune cells, microglia

## Abstract

**Simple Summary:**

The neuroinflammatory response is a rather complex event in spinal cord injury (SCI) and has the capacity to exacerbate cell damage but also to contribute to the repair of the injury. This complexity is thought to depend on a variety of inflammatory mediators, of which tumor necrosis factor (TNF) plays a key role. Evidence indicates that TNF can be both protective and detrimental in SCI. In the present study, we studied the temporal and cellular expression of TNF and its receptors after SCI in mice. We found TNF to be significantly increased in both the acute and the delayed phases after SCI, alongside a robust neuroinflammatory response. As we could verify some of our results in human postmortem tissue, our results imply that diminishing the detrimental immune signaling after SCI could also enhance recovery in humans.

**Abstract:**

Spinal cord injury (SCI) initiates detrimental cellular and molecular events that lead to acute and delayed neuroinflammation. Understanding the role of the inflammatory response in SCI requires insight into the temporal and cellular synthesis of inflammatory mediators. We subjected C57BL/6J mice to SCI and investigated inflammatory reactions. We examined activation, recruitment, and polarization of microglia and infiltrating immune cells, focusing specifically on tumor necrosis factor (TNF) and its receptors TNFR1 and TNFR2. In the acute phase, TNF expression increased in glial cells and neuron-like cells, followed by infiltrating immune cells. TNFR1 and TNFR2 levels increased in the delayed phase and were found preferentially on neurons and glial cells, respectively. The acute phase was dominated by the infiltration of granulocytes and macrophages. Microglial/macrophage expression of *Arg1* increased from 1–7 days after SCI, followed by an increase in *Itgam*, *Cx3cr1*, and *P2ry12*, which remained elevated throughout the study. By 21 and 28 days after SCI, the lesion core was populated by galectin-3^+^, CD68^+^, and CD11b^+^ microglia/macrophages, surrounded by a glial scar consisting of GFAP^+^ astrocytes. Findings were verified in postmortem tissue from individuals with SCI. Our findings support the consensus that future neuroprotective immunotherapies should aim to selectively neutralize detrimental immune signaling while sustaining pro-regenerative processes.

## 1. Introduction

Spinal cord injury (SCI) is a serious neurological condition with an unknown prevalence and estimated annual incidence of between 40 and 80 cases per million of population, according to the World Health Organization. SCI often leads to irreversible motor and sensory dysfunction below the level of injury. The mechanical impact to the spinal cord initiates a primary injury, which is followed by secondary degenerative processes. The secondary degeneration occurs due to detrimental cellular and molecular events, which include glutamate excitotoxicity, edema formation, and exacerbated neuroinflammation [1]. Neuroinflammatory processes are rapidly initiated after SCI and contribute to both injury and reparative processes [2]. Although the neuroinflammatory response is most pronounced in the early phases after SCI, it continues throughout the life of the affected individual [3]. Within minutes after injury, inflammatory mediators, such as cytokines, are released by resident cells located in the injured spinal cord and take part in the recruitment, activation, and polarization of immune cells [4].

One important inflammatory cytokine is the tumor necrosis factor (TNF), which plays a role in the initiation, maintenance, and resolution of inflammation [5]. It exists in two forms: a transmembrane-bound form (tmTNF) and a soluble form (solTNF). Both types of TNF signal through one of two receptors, TNFR1 and TNFR2, however, with different binding affinities, and the robust activation of TNFR2 requires binding of tmTNF [5,6]. Furthermore, the downstream signaling pathways of the two receptors differ, and the activation of TNFR1, especially, is associated with the increased expression of pro-inflammatory cytokines and activation of programmed cell death [7], whereas TNFR2 is involved in cell survival, proliferation, and remyelination [8,9].

Several studies have examined the cellular and temporal expression of TNF and its receptors in the acute phase after SCI [10,11,12,13,14,15,16,17]. However, only a few studies focused on clarifying TNF expression in the delayed phase after SCI [11,13,18]. Besides the initial acute increase in TNF levels, these studies suggest a second increase in *Tnf* gene expression in the delayed phase after SCI. Examining TNFR1 and TNFR2 expression in the delayed phase after SCI is, to our knowledge, yet to be elucidated. As elevated TNF levels induce the expression of numerous other inflammatory cytokines [19], studies have tried to clarify the role of TNF after SCI [20,21,22,23]. Studies using conventional and cell-specific conditional TNF or TNFR-knockout mice [20,24,25,26,27], and studies using anti-TNF therapy, [21,28,29] demonstrate that TNF exhibits both neuroprotective and neurodegenerative effects after SCI. In addition, we and others have shown that interfering with solTNF-TNFR1 signaling is beneficial after SCI [22,28].

Diminishing detrimental neuroinflammatory processes, such as the excessive production of pro-inflammatory cytokines, is considered a possible therapeutic strategy in individuals with SCI; therefore, more detailed knowledge on the temporal and cellular synthesis of inflammatory mediators is required. TNF is believed to be one of the most promising neuroinflammatory targets in SCI [17]; therefore, this study investigated the temporal and cellular source of TNF and its two receptors in the acute and delayed phases after SCI using a moderate contusive SCI model in C57BL/6J mice. The findings were verified in postmortem tissue and cerebrospinal fluid (CSF), derived from individuals with SCI. In addition, we evaluated the expression profile of selected glial-derived cytokines (IL-1 β, IL-6, IL-10, and CXCL1) and examined the polarization of microglia/macrophages by investigating temporal changes in microglial/macrophage specific genes (*Itgam*, *Cx3cr1*, *Trem2*, *Arg1*, *P2ry12*). Finally, we examined the activation, recruitment, and polarization of immune cells in the lesioned spinal cord.

## 2. Materials and Methods

### 2.1. CSF Collection

Human CSF samples were collected and stored at the Third Affiliated Hospital of Sun Yat-sen University in China. All individuals received a diagnosis of SCI based on clinical symptoms (ISNCSCI and ASIA score), electrophysiology, X-ray, and MRI analysis. Cases were divided into subacute (2 weeks–2 months after injury), early chronic (2–12 months after injury), and late chronic stages (>24 months after injury) (Table 1). SCI cases with complete or incomplete traumatic injury at cervical and thoracic levels were included in this study. One case had a lumbar SCI. Individuals with other neurological disorders or diabetes were excluded. CSF from healthy individuals was used as the control (Table 1), and samples were collected after overnight fasting and frozen at −80 °C. The study protocols ([2018]-02, [2018]-03, [2018]-04) were in accordance with guidelines for clinical studies approved by the Third Affiliated Hospital of Sun Yat-sen University review board.

### 2.2. Animals

Female C57BL/6J mice were purchased from Taconic A/S (Ry, Denmark) and transferred to the Biomedical Laboratory, University of Southern Denmark, where they were allowed to acclimatize for at least one week before surgery. TNF knockout (*Tnf^−/−^* [30]) mice were obtained by crossing heterozygous *Tnf^+/−^* mice, and the genotype was established using the following primers from DNA Technology A/S (Copenhagen, Denmark): *Tnf* common (5′-CCAGGAGGGAGAACAGA), *Tnf* mutant (5′-CGTTGGCTACCCGTGATATT), *Tnf* wt (5′-AGTGCCTCTTCTGCCAGTTC), *Lta*N forward (5′-GTCCAGCTCTTTTCCTCCCAAT), and *Lta*N reverse (5′-GTCCTTGAAGTCCCGGATACAC) as previously described [30,31]. All mice were group-housed with food and water ad libitum, with a 12 h light/dark cycle, and controlled temperature and humidity. Mice were cared for in accordance with the protocols and guidelines approved by the Danish Veterinary and Food Administration (J. numbers 2013–15–2934–00924 and 2019–15-0201–01615); experiments are reported in accordance with the ARRIVE guidelines, and all efforts were made to minimize pain and distress.

### 2.3. Induction of Spinal Cord Injury (SCI)

Mice were anesthetized with an intraperitoneal (i.p.) injection of a cocktail of ketamine (100 mg/kg, VEDCO, Saint Joseph, MO, USA) and xylazine (10 mg/kg, VEDCO). The ninth thoracic vertebra (T9) was identified based on anatomical landmarks [32], and the mice were laminectomized at T8–T10. Mice received a T9 contusion injury (75 Kdyn) using the Infinite Horizon Device (Precision Systems and Instrumentation, Brimstone, LN, USA) as previously described [28]. Sham mice were laminectomized only. After surgery, mice received a subcutaneous (s.c.) injection of isotonic saline to prevent dehydration, and for post-surgical analgesia, mice were treated with four s.c. injections of buprenorphine hydrochloride (0.001 mg/20 g body weight Temgesic, cat. no. 521634, Indivior Europe, North Chesterfield, VA, USA) at eight-hour intervals starting immediately after surgery. To prevent dehydration and infection, mice were supplemented with daily s.c. injections of isotonic saline and antibiotic gentamicin (40 mg/kg, Hexamycin, Sandoz, Copenhagen, Denmark) for the first 7 days after SCI. Mice were housed in individual cages in a recovery room at 25 °C with a 12 h light/dark cycle, until their wounds healed. Mice surviving more than 24 h after surgery were weighed at 1, 3, and 7 days after SCI and thereafter weekly. Bladders were emptied manually twice a day for the duration of the experiments. C57BL/6J mice were allowed to survive 1, 3, 6, 12, or 24 h (acute phase, n(SCI) = 18–23/group and n(sham) = 5/group) or 3, 7, 14, 21, or 28 days (delayed phase, n(SCI) = 15/group and n(sham) = 5/group) after surgery, and naïve mice were used as the controls (*n* = 20). *Tnf^−/−^* mice were allowed to survive 1, 3, 6, 12, or 24 h after SCI (*n* = 2/group). In total, one C57BL/6J mouse died during surgery.

### 2.4. Basso Mouse Scale (BMS)

Functional recovery after SCI was determined by scoring of the hind limb locomotor performance in the open field arena using the BMS scoring system and the BMS subscore system, with the latter used to quantify finer aspects of locomotion [33]. Under observer-blinded conditions, mice were evaluated over a 4 min period 1, 3, and 7 days after SCI and weekly thereafter for up to 28 days. Before surgery, mice were handled and pre-trained in the open field to assure normal locomotion and to prevent fear and/or stress behaviors that could bias the locomotor assessment. Routinely, mice with a BMS score above 1 on the day after surgery are excluded; in the present study, however, no mice scored above 1.

### 2.5. Human SCI Tissue

Paraffin-embedded postmortem human spinal cord samples (Table 2) were obtained from the Miami Project Human Core Bank at the University of Miami Miller School of Medicine managed by Alexander Marcillo, M.D. and Yan Shi, M.S.

### 2.6. Tissue Processing

Mice were deeply anesthetized with an overdose of pentobarbital (200 mg/mL) containing lidocaine (20 mg/mL) (Glostrup Apotek, Glostrup, Denmark) and transcardially perfused through the left ventricle. For reverse transcription quantitative polymerase chain reaction (RT-qPCR), in situ hybridization, protein analysis, and flow cytometry, mice were perfused with ice-cold diethyl pyrocarbonate-treated (DEPC, Sigma-Aldrich, cat. no. D5758, Soeborg, Denmark) phosphate-buffered saline (PBS, pH 7.4, Sigma-Aldrich, cat. no. P4417, Soeborg, Denmark). For immunohistochemistry and immunofluorescence staining, mice were perfused with ice-cold 4% paraformaldehyde (PFA, Sigma-Aldrich, cat. no. 158127, Soeborg, Denmark) diluted in PBS.

For RT-qPCR and protein analysis, 1 cm of spinal cord centered on the lesion area (SCI samples), or spinal cord tissue taken from the equivalent region (sham and naïve samples), was quickly removed, snap-frozen on dry ice, and stored at −80 °C until further processing.

Spinal cord segments (1 cm centered on the lesion), to be used for in situ hybridization, immunohistochemistry, and immunofluorescence staining, were quickly removed. For in situ hybridization, segments were immediately embedded in Tissue-Tek OCT compound (Leica, cat. no. 14020218926, Broendby, Denmark) and snap-frozen in gaseous CO_2_. Spinal cord segments used for immunohistochemistry and immunofluorescence staining were stored in PFA for 45 min, hereafter in 20% sucrose (Sigma-Aldrich, cat. no. S7903, Soeborg, Denmark) in 0.15 M Sorensen’s phosphate buffer overnight (o.n), and the next day embedded and snap-frozen in Tissue-Tek compound. Spinal cords were then cut into 20 µm thick parallel tissue sections using a cryostat, collected on SuperFrost Plus slides (Thermo Fisher Scientific, cat. no. 10149870, Roskilde, Denmark), and stored at −20 °C (immunostaining) or −80 °C (in situ hybridization) until further processing.

For flow cytometry, spinal cord tissue containing the lesion area (1 cm centered on the lesion) and peri-lesion area (tissue 0.5 cm distal and 0.5 cm proximal to the lesion was pooled to represent peri-lesion tissue), or spinal cord tissue taken from the equivalent regions (naïve samples), was quickly removed and placed in cold RPMI (Gibco, cat. no. 21875–042, Roskilde, Denmark) containing 10% fetal bovine serum (FBS, VWR, cat. no. S1810, Soeborg, Denmark). Samples were homogenized through a 70 μm filter (AH Diagnostics, cat. no. 352350, Aarhus, Denmark) and further processed for flow cytometry.

### 2.7. Gene Analysis

RNA extraction: Total RNA was extracted from mice that survived 1, 3, 6, 12, and 24 h, 3, 7, 14, and 28 days after SCI, as well as from naïve controls (*n* = 5/group) using TRIzol Reagent (Invitrogen, cat. no. 15596018, Roskilde, Denmark) according to the manufacturer’s protocol. Briefly, samples were homogenized with the appropriate amount of TRIzol Reagent, and chloroform extraction (Sigma-Aldrich cat. no. C2432, Soeborg, Denmark) was performed followed by isopropanol precipitation (Sigma-Aldrich, cat. no. I9030, Soeborg, Denmark). The RNA was washed with 75% ethanol (absolute ethanol in nuclease-free water, VWR, cat. no. 20821.365), and purified RNA was dissolved in nuclease-free water (Thermo Scientific, cat. no. R0582). Concentrations and purity were checked using a Thermo Scientific NanoDrop One spectrophotometer.

cDNA synthesis: Two µg RNA was reverse-transcribed with the High-Capacity cDNA Reverse Transcription kit from Applied Biosystems (Thermo Fisher, cat. no. 4368814, Roskilde, Denmark). A 2× reverse transcription (RT) Master mix was made, consisting of a 10X RT Buffer, 25X dNTP mix (100 mM), 10 RT Random Primers, MultiScribe Reverse Transcriptase, nuclease-free water, and equal amounts of RNA sample; the 2× RT Master mix was synthesized using an MJ Research PTC-225 Gradient Thermal Cycler from Marshall Scientific. Reverse transcription cycle conditions were as follows: 25 °C for 10 min, 37 °C for 120 min, 85 °C for 5 min, and then cooled to 4 °C. Samples were diluted to 50 ng/µL and stored at −20 °C until further processing.

RT-qPCR: Investigation of *Tnf*, *Tnfrsf1a*, *Tnfrsf1b*, *Il1b*, *Il6*, *Il10*, *Cxcl1*, Integrin subunit alpha M (*Itgam*), C-x3-c motif chemokine receptor 1 (*Cx3cr1*), triggering receptor expressed on myeloid cells 2 (*Trem2*), purinergic receptor P2Y (*P2ry12*), arginase 1 (*Arg1*), and hypoxanthine-guanin phosphoribosyltransferase 1 (*Hprt1*) mRNA expression was performed with Maxima SYBR Green (ThermoFisher Scientific, cat. no. KO223, Roskilde, Denmark) detection and carried out using a CFX Connect Real-Time PCR Detection System from Bio-Rad. Primers were designed with NCBI’s nucleotide database and primer designing tool, aimed to target exon–exon junctions whenever possible, checked for self-complementarity with an Oligo calculator [34], and purchased from TAG Copenhagen (Copenhagen, Denmark). Primer sequences are listed in Table 3. The RT-qPCR reaction was performed in a 12.5 µL volume, containing 1× Maxima SYBR Green, 50 ng of template cDNA, and 600 nM forward and reverse primers. Thermal cycling conditions were as follows: 95 °C for 10 min to separate the cDNA, followed by further denaturation for 15 s, whereafter the temperature was lowered to the optimal annealing temperature for each gene (see Appendix A) for 30 s and then raised again to 72 °C for 30 s. This was caried out for 40 cycles, except for *Il10* (45 cycles, Appendix A). Finally, the samples were heated to generate a melting curve (Appendix A). A 4-fold standard curve and a calibrator were prepared from a mixture of aliquots from all experimental samples and used on every assay. “No template” and “no reverse transcriptase” controls were included as negative controls. All samples and standards were tested in triplicate, the calibrator was applied to six wells, and samples from different time points were randomly distributed across the assays. Amplification of a single desired product was confirmed by the presence of only one melting curve. Relative transcript levels were calculated using the *Pfaffl* method [35], primer efficiencies were accepted within the range of 100 ± 5% (Appendix A), and data were normalized to the reference gene *Hprt1*.

### 2.8. In Situ Hybridization for Tnf mRNA

In situ hybridization for *Tnf* mRNA was performed using a mixture of two alkaline phosphatase (AP)-labeled oligo DNA probes (3 pmol/mL) on tissue sections from C57BL/6J mice surviving 1, 3, 6, 12, and 24 h after SCI, in addition to naïve controls (*n* = 3/group). The following probes were purchased from DNA Technology (Copenhagen, Denmark): *Tnf* probes: 5′ CGTAGTCGGGGCAGCCTTGTCCCTTGAA 3′ (GC content 60.7%, Tm 67.8 °C) and 5′ CTTGACGGCAGAGAGGAGGTTGACTTTC 3′ (GC content 53.6%, Tm 62.3 °C); glyceraldehyde 3-phosphate dehydrogenase (*Gapdh*) probe: 5′ CCTGCTTCACCACCTTCTTGATGTCA 3′ (GC content 50%, Tm = 60.2 °C). The hybridization was performed on 20 μm ethanol-fixed spinal cord sections using protocols previously described [36,37]. The hybridization signal was developed using an AP buffer containing 5-bromo-4-chloro-3-indolyl phosphate (Sigma-Aldrich, cat. no. B8503, Soeborg, Denmark) and nitroblue tetrazolium (Sigma-Aldrich, cat. no. N6876, Soeborg, Denmark). The specificity of the hybridization was documented by (1) the abolishment of the hybridization signal when hybridizing RNase A-digested sections, (2) hybridizing sections with 100-fold excess of the unlabeled probe mixture, or (3) the absence of signal in sections incubated with buffer only. Parallel sections were hybridized for the widely expressed *Gapdh* mRNA to ensure overall suitability of the tissue for hybridization.

### 2.9. Protein Purification

Spinal cord tissue segments from naïve mice or mice surviving 1, 3, 6, 12, or 24 h and 3, 7, 14, 21, or 28 days after SCI (*n* = 5/group) were thawed on ice, sonicated in lysis buffer (150 mM sodium chloride (Sigma-Aldrich, cat. no. 1064041000), 20 mM Tris, 1 mM Ethylene Diamine Tetra Acetate (EDTA, Sigma-Aldrich, cat. no. E9884, Soeborg, Denmark), 1 mM ethylene glycol tetraacetic acid (EGTA, Sigma-Aldrich cat. no. E4378, Soeborg, Denmark), 1% Triton-X-100, a cocktail of phosphatase and proteinase inhibitors (Sigma-Aldrich, P5726 and Sigma-Aldrich, P0044, Soeborg, Denmark), and a cOmplete™ Mini EDTA-Free Tablet (Roche, 11836170001), pH 7.5). Samples were left shaking on ice at 4 °C for 30 min, centrifuged at 14,000× *g* at 4 °C for 20 min, and finally the supernatants were stored at −80 °C until further analysis. The protein concentration was determined using the Pierce BCA Protein Assay Kit (Thermo Fischer Scientific, cat. no. 23235, Roskilde, Denmark) according to the manufacturer’s protocol.

### 2.10. Electrochemiluminescence Analysis

TNF, IL-1β, IL-6, IL-10, CXCL1, TNFR1, and TNFR2 protein levels were measured in tissue homogenates from SCI or sham mice surviving 1, 3, 6, 12, or 24 h (acute phase) and 3, 7, 14, or 28 days (delayed phase) survival, as well as from naïve controls (*n* = 5/group), using custom made MSD Mouse Pro-inflammatory V-PLEX (Mesoscale Discovery, Rockville, MD, USA, cat. no. K152BIC (acute) and K152AOH-2 (delayed)), Ultra-sensitive TNFRI (Mesoscale Discovery, cat. no. K152BIC (acute) and K1510VK-2 (delayed)), and TNFRII (Mesoscale Discovery, cat. no. K152BJC (acute) and K150ZSR-2 (delayed)) kits, as previously described [26]. Analysis of tissue derived from mice surviving 3, 7, 14, and 28 days (delayed phase) was performed separately from tissue derived from mice surviving 1, 3, 6, 12, and 24 h (acute phase) and, thus, they were analyzed as two separate experiments. Samples were diluted in Diluent 41, run in duplex on a SECTOR Imager 6000 Plate Reader (Mesoscale Discovery), and analyzed using MSD Discovery Workbench software. Samples with coefficient of variation (CV) values >25% in individual analyses were excluded. The lower limit of detection (LLOD) was a calculated concentration based on a signal 2.5 standard deviations (SD) above the blank (zero) calibrator. For protein levels below LLOD, a value of 0.5 LLOD was used for statistical analysis. LLOD values for acute experiments; IL-1β (0.19–0.22 pg/mL), IL-10 (0.81–1.34 pg/mL), CXCL1 (0.14–0.23 pg/mL), TNF (0.22–0.77 pg/mL), IL-6 (1.70–4.04 pg/mL), TNFR1 (0.57–0.61 pg/mL), and TNFR2 (15.00–35.90 pg/mL). LLOD values for delayed experiments; IL-1β (0.09 pg/mL), IL-10 (0.61 pg/mL), CXCL1 (0.13 pg/mL), TNF (0.15 pg/mL), IL-6 (1.57 pg/mL), TNFR1 (0.16 pg/mL), and TNFR2 (0.73 pg/mL).

### 2.11. CSF ELISA

ELISA tests on human CSF were performed at the Third Affiliated Hospital of Sun Yat-sen University in China. SolTNF (ab181421) and solTNFR1 (ab209881) were quantitatively measured using commercially available ELISA kits (Abcam, Cambridge, UK) according to the manufacturer’s instructions. Before testing, CSF samples were centrifuged at 2000× *g* for 10 min to remove debris. Supernatants were collected for further analysis.

### 2.12. Immunohistochemistry for TNF

Visualizing the TNF protein in spinal cord tissue sections from C57BL/6 (*n* = 3/group) and *Tnf^−/−^* (*n* = 2/group) mice surviving 1, 3, 6, 12, and 24 h after SCI and naïve controls (*n* = 3) was performed using a two-step immunohistochemical protocol with an AP-conjugated secondary antibody as previously described in detail [37]. In short, sections were air-dried and fixed in 4% PFA for 10 min. Sections were then rinsed in Tris-buffered saline (TBS, pH 7.4) for 15 min, TBS + 0.1% Triton (Merck, cat. no. X100, Soeborg, Denmark) 3 × 15 min, and incubated with 10% FBS in TBS for 30 min at room temperature. Thereafter, sections were incubated with polyclonal rabbit anti-mouse TNF antibody (Table 4) diluted at 1:200 in 10% FBS in TBS for 1 h at room temperature, followed by 48 h at 4 °C. Next, sections were rinsed 3 × 15 min in TBS + 0.1% Triton and incubated with a secondary AP-conjugated antibody to rabbit IgG (Jackson ImmunoResearch, cat. no. 111–055-003, Cambridgeshire, UK) diluted at 1:200 in 10% FBS in TBS for 1 h at room temperature. The antigen–antibody complex was visualized using the AP developer used for in situ hybridization containing 1 mol/L Levamisole (Sigma-Aldrich, cat. no. PHR1798, Soeborg, Denmark). Finally, the development was arrested in distilled water, and the sections were cover slipped in Aquatex (Sigma-Aldrich, cat. no. 1.085.620.050, Soeborg, Denmark).

Control reactions for antibody specificity were performed on parallel sections by (1) substitution of the primary antibody with rabbit serum (Dako, cat. no. X0902, Glostrup, Denmark), (2) omission of the primary antibody in the protocol, and (3) inclusion of sections from *Tnf^−/−^* mice. *Tnf^−/−^* mice are known to be devoid of functional TNF protein [31]. Parallel sections incubated with polyclonal rabbit anti-glial fibrillary acidic protein (GFAP, Table 4) diluted at 1:4000 were included for the overall control of the immunohistochemical reaction.

### 2.13. Immunofluorescence Staining

*Mouse tissue*: Double immunofluorescence staining for TNF, TNFR1, or TNFR2 with cell-specific markers was performed on 20-µm-thick, parallel cryostat tissue sections from mice surviving 3 h, 21 days, or 28 days after SCI. Double labelling for GFAP with CD11b, CD68, or Iba1 was performed on tissue sections from mice surviving 21 or 28 days after SCI (*n* = 2–3/group). Sections were blocked with 10% FBS in TBS for 30 min, incubated o.n. with primary antibodies (Table 4), rinsed in TBS, and incubated with fluorescently labelled secondary antibodies for 2 h at room temperature (Table 4). For visualization of astrocytes, Cy3- or 488-conjugated anti-GFAP antibodies (Table 4) were applied for 1 h at room temperature. Sections were rinsed in TBS containing 4′,6-diamidino-2-phenylindole (DAPI, 1:1000, Sigma-Aldrich, cat. no. D9542, Soeborg, Denmark) to visualize nuclei and mounted with Aquatex.

*Human tissue*: Human postmortem spinal cord tissue was formalin-fixed, embedded in paraffin, and cut into 10-μm-thick sections on a microtome. Tissue sections were deparaffinized in xylene and rehydrated in graded series of ethanol (99%, 96%, 70%, 50%), immersed in water, and washed in PBS before heat-induced epitope retrieval in citrate buffer (10 mM citrate, pH 6). Next, sections were rinsed in PBS and bleached using the Autofluorescence Eliminator Reagent (Millipore, cat. no. 2160, Soeborg, Denmark) according to the manufacturer’s guidelines [38]. The sections were then rinsed in PBS followed by TBS and TBS + 0.1% triton before blocking in 10% FBS in TBS for 30 min at room temperature. The sections were incubated o.n. with primary antibodies diluted in 10% FBS in TBS (Table 4). The following day, the sections were rinsed in TBS + 0.1% triton, and incubated for 2 h with secondary antibodies (Table 4) diluted in 10% FCS in TBS at room temperature, protected from light. Finally, sections were rinsed in TBS before mounting with ProLong Gold Antifade Reagent with DAPI (Sigma-Aldrich, cat. no. 10236276001, Soeborg, Denmark).

### 2.14. Flow Cytometry

Samples from mice surviving 3 and 24 h and 14, 21, and 28 days survival after SCI as well as naïve controls (*n* = 5/group) were processed for flow cytometry using a FACSCalibur flow cytometer and data analyzed using FACSuite software, as previously described [39]. Tissue from individual mice was processed individually, and approximately 10^6^ events were acquired per sample using forward scatter (FSC) and side scatter (SSC). Microglia (CD11b^+^CD45^dim^), macrophages (CD11b^+^CD45^high^Ly6C^high^Ly6G^−^), and granulocytes (CD11b^+^CD45^high^Ly6C^+^Ly6G^+^) were identified as previously described [26]. Prior to fixation, cells were stained for live/dead cells using Fixable Viability Dye eFluoro 506 (Thermo Fischer, cat. no. 65–0866-18) diluted in PBS. Positive staining was determined based on isotype controls and the respective fluorescent minus one (FMO) control [25]. Antibodies were directly conjugated with fluorochromes (Table 5). The mean fluorescence intensity (MFI) was calculated as the geometric mean of each population in the CD45 and CD11b positive gates.

### 2.15. Statistical Analysis

Comparisons were performed using repeated measures (RM) or regular two-way analysis of variance (ANOVA) followed by Sidak’s post hoc analysis, ordinary one-way ANOVA followed by Dunnet’s post hoc analysis, or by Student’s *t*-test. Correlation analyses were performed using the nonparametric Spearman correlation test. Outliers were identified using ROUT with a False Discovery Rate (FDR) of 1%. Analyses were performed using Prism 4.0b software for Macintosh, (GraphPad Software, San Diego, CA, USA). Statistical significance was established for *p* < 0.05. Data are presented as mean ± standard error of mean (SEM), percentages, or as mean with interquartile range (IQR 25–75%).

## 3. Results

### 3.1. SCI Leads to Significant Changes in Locomotor Function

The recovery of hind limb function after a moderate contusive SCI was evaluated using the BMS score (Figure 1a) and the BMS subscore (Figure 1b). Mice exhibited immediate paraplegia with no hind limb movement after induction of SCI. Mice receiving a laminectomy (sham) only displayed minor impairment 1 day after surgery but displayed normal motor function from 3 days and onwards after surgery, as demonstrated by a normal BMS (Figure 1a) and normal BMS subscore (Figure 1b). Mice with SCI started to display improved hind limb function from day 7 and onwards but exhibited significant motor dysfunction compared to sham mice throughout the experiment (Figure 1a). Improved locomotion in SCI mice was also detected from day 14 using the BMS subscore (Figure 1b). Mice subjected to SCI experienced significant weight loss within the first 3 days after SCI, whereafter they started to regain weight (Figure 1c). Sham mice did not experience any weight loss (Figure 1c).

### 3.2. SCI Results in Glial Scar Formation

Immunofluorescent staining for the astrocyte-specific marker GFAP and microglial/macrophage-specific markers CD11b, CD68, and Iba1 at 21 and 28 days after SCI showed that activated GFAP^+^ astrocytes formed a dense glial scar at the injury border surrounding a core lesion consisting of activated, phagocytic microglia/macrophages (Figure 1d). CD11b^+^, CD68^+^, and Iba1^+^ cells were also located in the peri-lesion areas surrounding the core lesion area although not to the same extent (Figure 1d).

### 3.3. Tnf mRNA Synthesis Increases in the Acute Phase after SCI

We used in situ hybridization to determine the topology of cells expressing *Tnf* mRNA in the acute phase after SCI. No *Tnf* mRNA was detected under naïve conditions (Figure 2a). Already after 1 h, *Tnf* mRNA expression was detected in glial-like cells located in the white matter of the posterior funiculi (Figure 2b) and also in neuron-like cells located in the grey matter of the dorsal horn (Figure 2c). At 3 h after SCI, a high number of *Tnf* mRNA-expressing glial-like cells were scattered throughout the lesioned posterior funiculi white matter (Figure 2d) as well as in the dorsal and ventral horns (Figure 3e, shown for the ventral horn only). *Tnf* mRNA^+^ cells in the white matter had a glial-like morphology (Figure 3f). At 6 h (Figure 2g), 12 h (Figure 2h), and 24 h (Figure 2i) after SCI, *Tnf* mRNA^+^ glial-like cells were still found in the posterior and lateral funiculi although fewer in number than at 3 h. *Gapdh* mRNA expression was used as control, confirming the suitability of the tissue for in situ hybridization (Figure 2j). Buffer (Figure 2k) and RNase (Figure 2l) controls were devoid of signal.

The in situ hybridization data were supported by RT-qPCR analysis of the temporal expression of *Tnf* mRNA in spinal cord tissue after SCI (Figure 2m). *Tnf* mRNA levels increased rapidly after SCI, with the highest expression detected at 1 and 3 h after SCI compared to naïve, and levels were still elevated at 6 h after SCI. A small, but significant, increase was also observed 7 days after SCI (Figure 2m).

The gene expression of the two TNF receptors, *Tnfrsf1a* and *Tnfrsf1b*, was also measured using RT-qPCR analysis (Figure 2n,o). *Tnfrsf1a* mRNA expression steadily increased from 6 h after SCI, reaching peak levels at 14 days after SCI, whereafter it started to decline, although it was still significantly elevated at 28 days compared to naïve (Figure 2n).

*Tnfsf1b* mRNA expression was significantly increased 3 h after injury and quickly decreased thereafter (Figure 2o). A second increase was observed 7 days after SCI; however, this increase did not quite reach statistical significance (*p* = 0.08).

### 3.4. TNF Is Increased on Glial Cells after SCI

To investigate TNF protein levels after SCI, we performed electrochemiluminescence multiplex analysis on spinal cord tissue from mice subjected to SCI and compared to naïve and sham mice (Figure 3a,b). TNF levels increased significantly over time in SCI mice compared to naïve mice. In line with our in situ hybridization and RT-qPCR analysis, TNF levels peaked 3 h after SCI and was still elevated at 6 h, compared to sham and naïve mice (Figure 3a). In the more delayed phase, TNF levels were significantly increased at 3 and 7 days after SCI, compared to sham and naïve mice (Figure 3b). Using immunohistochemistry, TNF immunoreactivity was demonstrated in scattered cells located in the dorsal horns and posterior and lateral funiculi, as well as around blood vessels 1 h after SCI (Figure 3c,d). TNF immunoreactivity was intensified in the posterior part of the damaged spinal cord 3 h after SCI (Figure 3e,f), at which time point *Tnf* mRNA (Figure 3g) and TNF protein (Figure 3h) expression was mostly confined to GFAP^−^ cells although a few GFAP^+^ astrocytes also co-expressed TNF (insert in Figure 3h). At 6 h after SCI, TNF immunoreactivity was localized throughout the damaged spinal cord, especially around blood vessels (Figure 3i–k). At 24 h, TNF expression was scarcely distributed throughout the spinal cord, with the overall TNF immunoreactivity decreasing at this time point compared to earlier time points (Figure 3l,m). Immunofluorescence double labeling demonstrated that TNF co-localized to CD11b^+^ immune cells located within the lesion and in the peri-lesion area (Figure 3n) at 21 and 28 days after SCI. Serum (Figure 3q) and buffer (Figure 3r) controls were devoid of staining. Immunohistochemical staining for the abundant astroglial marker GFAP was used as a positive control for the immunohistochemical procedure (Figure 3s). Tissue sections from *Tnf^−/−^* mice subjected to SCI were devoid of specific TNF immunoreactivity (Figure 3t), just as double fluorescent staining for GFAP and TNF in *Tnf^−/−^* mice subjected to SCI were devoid of specific TNF staining (Figure 3u).

### 3.5. TNFR1 and TNFR2 Expression Increases in the Lesioned Spinal Cord

We analyzed the temporal and cellular expression of TNFR1 and TNFR2 in spinal cord tissue from naïve and sham mice, as well as mice that had survived 1, 3, 6, 12, and 24 h and 3, 7, 14, and 28 days after SCI (Figure 4). TNFR1 levels were found to increase significantly from 24 h after SCI and onwards (Figure 4a,b). Double immunofluorescent staining showed that TNFR1 expression was absent within the core of the lesion site, where microtubule associated protein 2^+^ (MAP2^+^) neurons were also absent (Figure 4c). TNFR1 expression was upregulated in areas of MAP2^+^ degenerating neurons located in the grey matter of the peri-lesion area (Figure 4d), and its expression co-localized to ascending and descending fiber tracts more distant from the lesion area at 21 days after SCI (Figure 4e). Additionally, at 28 days, TNFR1 expression was upregulated in areas near the lesion site, alongside MAP2^+^ degenerating neurons (Figure 4f). TNFR1 expression was found to co-localize to the soma and fibers of MAP2^+^ cells (Figure 4g, shown for 28 days only). CD68^+^ microglia/macrophages aligned along the damaged ascending and descending fiber tracts of the white matter, where TNFR1 expression was also localized (Figure 4h, shown for 21 days only).

TNFR2 levels were not significantly altered in the acute phase after SCI (Figure 4i), but levels increased significantly in the more delayed phase, i.e., from 3 days after SCI and onwards (Figure 4j). Double immunofluorescence staining demonstrated that at 21 and 28 days after SCI, TNFR2 co-localized to GFAP^+^ astrocytes forming the glial scar and CD11b^+^ microglia (Figure 4k–n), but to some extent also to cells, possibly infiltrating macrophages, located in the core of the lesion (Figure 4m,n). At 21 days after SCI, immunofluorescent double labeling of CD68^+^ microglia/macrophages and TNFR2 expression (Figure 4o–q) showed that TNFR2 expression was absent in most CD68^+^ cells (Figure 4o), although a minority of the cells co-expressed TNFR2 (Figure 4p,q).

### 3.6. SCI Results in Increased Levels of Inflammatory Cytokines

To assess the temporal expression of selected inflammatory cytokines known to be important after SCI [4,40], we analyzed gene and protein expression levels of IL-1β, IL-6, IL-10, and CXCL1 after SCI (Figure 5). We found that *Il1b* mRNA levels increased significantly from 3–12 h after SCI, compared to naïve conditions (Figure 5a). This increase was followed by a transient increase in IL-β levels at 6 and 12 h after SCI (Figure 5b). A second increase in IL-1β levels could be detected in the delayed phase after SCI, from day 7 and onwards, with the highest expression on day 14 (Figure 5c). *Il6* mRNA levels significantly increased at 6 and 12 h after SCI (Figure 5d) and was paralleled by a transient increase in IL-6 levels (Figure 5e). A second peak in IL-6 levels was found 3 days after SCI (Figure 5f). *Il10* mRNA levels increased rapidly at 1 and 3 h after SCI whereafter they decreased again (Figure 5g). IL-10 levels transiently increased 6 h after SCI (Figure 5h) and then returned to baseline levels (Figure 5h–i). *Cxcl1* mRNA levels did not change significantly after SCI, but there was a trend towards an increase at 6 h (*p* = 0.09, Figure 5j). In contrast, CXCL1 levels increased transiently at 6 and 12 h after SCI (Figure 5k), and again from day 3 and onwards (Figure 5l).

### 3.7. SCI Results in Microglial Activation and Immune Cell Infiltration into Spinal Cord

We performed flow cytometry analysis to estimate microglial and infiltrating immune cell populations in the spinal cord of naïve mice and mice surviving 3 and 24 h as well as 14, 21, and 28 days after SCI (Figure 6). We gated only live cells and included CD11b^+^CD45^dim^ microglia and infiltrating CD11b^+^CD45^high^ leukocytes, which were further gated into Ly6G^+^Ly6C^+^ granulocytes and Ly6G^−^Ly6C^+^ monocytes (Figure 6a). We estimated the total number of microglia (Figure 6b–c) and infiltrating leukocytes (Figure 6d–e), as well as the number of macrophages (Figure 6f–g) and granulocytes (Figure 6h–i), in the lesion (open bars) and peri-lesion (checkered bars) areas. We found that microglial cell numbers increased significantly in the lesion compared to the peri-lesion area 3 h after SCI and stayed high in both the lesion and peri-lesion areas at 24 h (Figure 6b). In the chronic phase after SCI, microglial numbers were comparable between the lesion and peri-lesion area, except for 21 days after SCI, where microglial numbers were increased in the lesion area compared to the peri-lesion area (Figure 6c) The total number of infiltrating leukocytes increased significantly within the lesion area 24 h after SCI (Figure 6d), with both significantly increased numbers of macrophages (Figure 6f) and granulocytes (Figure 6h) located within the lesion area. In the chronic phase after SCI the number of infiltrating leukocytes (Figure 6e), including macrophages (Figure 6g) and granulocytes (Figure 6i), was comparable between the lesion and peri-lesion sites. Changes in the percentages of the cell populations after SCI can be found in Appendix A.

MFI for CD11b (Figure 6j–o) and MFI for CD45 (Figure 6p–u) were significantly increased on microglia (Figure 6j,p) and macrophages (Figure 6l,r) located within the lesion area compared to the peri-lesion area, 24 h after SCI, just as MFI for CD11b and CD45 on microglia was increased in the lesion area compared to the peri-lesion area 28 days after SCI (Figure 6k,q). In the chronic phase after SCI, MFI for CD11b and MFI for CD45 on macrophages (Figure 6m,s) did not differ between the lesion and peri-lesion areas. MFI for CD11b and CD45 on granulocytes (Figure 6h,k) were already significantly increased in the lesion area 3 h after SCI and remained increased at 24 h, compared to the peri-lesion area. No differences were observed in the chronic phase after SCI (Figure 6o,u).

We further characterized microglial/macrophage responses in the spinal cord after SCI using RT-qPCR (Figure 7a–e). We found that *Itgam* (Figure 7a) and *Cx3cr1* (Figure 7b) mRNA levels increased significantly at day 3 and remained elevated until 28 days, compared to naïve controls. *Trem2* mRNA levels increased transiently 7 days after SCI, compared to naïve conditions (Figure 7c). *Arg1* mRNA levels increased significantly 24 h after SCI and remained elevated until day 7, after which they declined (Figure 7d). The gene expression for the purinergic receptor *P2ry12* was significantly elevated 3 days after SCI and remained elevated throughout the experiment (Figure 7e).

To evaluate the presence and location of activated microglia/macrophages in the injured spinal cord, we performed double immunofluorescence staining for microglial and macrophage-specific calcium-binding protein (Iba1), together with either galectin-3 (Gal3), a marker for microglial/macrophage activation [41] (upper panel in Figure 7f), the phagocytic lysosomal marker CD68 (middle panel in Figure 7f), or the general leukocyte and microglial marker CD11b (lower panel in Figure 7f) at 21 and 28 days after SCI. At both time points, Gal3, CD68, and CD11b expression was confined to CD11b^+^ cells located in the core of the lesion. In the peri-lesion area, only a subpopulation of Iba1^+^ cells co-expressed Gal3, CD68, or CD11b, demonstrating that different subsets of microglia express different markers in the peri-lesion area after SCI.

### 3.8. The Cellular Source of TNF and Its Receptors in Human Traumatic Spinal Cord Injury

Consistent with our mouse studies, we observed that TNF co-localized to a subset of Iba1^+^ microglia/macrophages (Figure 8a,b) and to a minority of CD68^+^ phagocytes (Figure 8c) in postmortem tissue derived from individuals with SCI. TNF did not co-localize to GFAP^+^ astrocytes (Figure 8d). TNFR1 co-localized to NF-L^+^ proximal dendrites (Figure 8e), and TNFR2 to GFAP^+^ astrocytes (Figure 8f) as well as Iba1^+^ microglia (Figure 8g), whereas TNFR2 did not co-localize to NF-L^+^ neurons (Figure 8h). IL-1β was abundant in the lesioned spinal cord (Figure 8i,j), but only a minority was expressed by Iba1^+^ microglia (Figure 8j). Similar to our mouse studies (Figure 7b), we observed CD68^+^Iba1^+^ phagocytic cells throughout the lesioned spinal cord (Figure 8k,l).

Using ELISA analysis, we observed no significant changes in CSF TNF levels between controls and individuals with sub-acute, early chronic, or late chronic SCI (Figure 8k). We observed no correlation between CSF TNF levels and age (Spearman ρ = 0.05, *p* = 0.84). In contrast, TNFR1 levels were significantly upregulated in individuals with SCI in the acute phase after SCI (Figure 8l). We observed no correlation between CSF TNFR1 levels and age (Spearman ρ = 0.16, *p* = 0.38).

## 4. Discussion

In the present study, we investigated neuroinflammatory responses and determined the temporal expression and cellular sources of TNF and its receptors, TNFR1 and TNFR2, in the acute and delayed phases after SCI.

It is well-known that SCI triggers a well-characterized innate cellular immune response initiated by microglia and amplified by peripheral myeloid cells, mainly neutrophils and monocytes, which migrate to the injury site [3]. By 3 days, most glial cells, including astrocytes and microglia, are at the peak of their proliferative state, resulting in the recovery of cell numbers and initiation of astroglial gliosis [42]. At the same time, monocytes differentiate into macrophages [43] and by 7 days, macrophages have reached their peak. In the present study, we found a rapid and transient increase in TNF expression between 1 and 6 h after SCI, followed by a more delayed increase between 3 and 14 days after SCI. The rapid increase in TNF levels implies that TNF is an acute driver of neuroinflammation after SCI, and this is supported by others [10,12,13,14,44,45,46]. Early after SCI, TNF was expressed within or just near the lesion, and a few *Tnf* mRNA+ cells were also detected near blood vessels, indicating that as well as modulating neuroinflammation locally, TNF also participates in signaling to the periphery [11,13]. Combined in situ hybridization and immunofluorescent staining for astrocytes co-localized *Tnf* mRNA to a few GFAP^+^ astrocytes 3 h after SCI, but *Tnf* mRNA^+^ neuron- and microglial-like cells were also present. Immunohistochemical analysis revealed that TNF was expressed by cells located in proximity to vessels, suggesting that infiltrating immune cells are major producers of TNF in the acute phase after SCI. These findings are supported by previous reports that also demonstrate TNF expression by neurons, microglia, and astrocytes, in addition to oligodendrocytes in the acute phase after SCI [12,13,14,17,47]. In the present study, we extend previous findings of increased *Tnf* mRNA levels in the delayed phase after SCI [13] by showing increased TNF expression from 3 to 14 days after SCI, correlating with the time points of peripheral immune cells infiltration into the injured spinal cord [40]. By 21 and 28 days after SCI, TNF was expressed mainly by CD11b^+^ cells, i.e., microglia and possibly infiltrating macrophages [20]. In the human spinal cord, we found TNF to be expressed mainly by Iba1^+^ cells, i.e., microglia and infiltrating macrophages, and to a lesser extent by CD68^+^ cells, presumably phagocytosing macrophages, demonstrating the translational relevance of our mouse studies. Speculation has been made as to whether the early increase in TNF expression exhibits the more detrimental role of TNF after SCI [47,48,49], while the secondary increase promotes neuroprotection and tissue healing [23]. However, blocking TNF with etanercept, a non-selective TNF inhibitor of both tmTNF and solTNF, 14 days after SCI did not affect functional recovery after SCI [18], and we recently showed that myeloid-derived TNF (e.g., macrophages and granulocytes), which are main contributors of TNF in the delayed phase [20], play a detrimental role in SCI [26], thus, leaving the assumption unresolved so far.

TNF mediates its signaling through its two receptors, TNFR1 and TNFR2 [50], and the upregulation of TNFR1 and TNFR2 by neurons, oligodendrocytes, and astrocytes, but not on resting microglia, have been reported in the acute phase after SCI [16]. In the present study, we demonstrated significantly increased *Tnfrsf1a* mRNA levels from 6 h after SCI, while TNFR1 levels increased significantly from 24 h and onwards after SCI. TNFR1 primarily co-localized to the soma of MAP2^+^ degenerating neurons located in the peri-lesion grey matter and to ascending and descending white matter fiber tracts. This was verified in human postmortem tissue, where TNFR1 co-localized to NF-L^+^ proximal dendrites in the injured spinal cord. TNFR1 is associated with neurodegeneration [51], and neuronal TNFR1 has been shown to enhance demyelination and exacerbate microglial inflammation [52]. Thus, our findings suggest that chronic inflammation after SCI may be sustained or promoted by increased TNFR1 expression. Additionally, TNFR2 levels increased significantly in the delayed phase after SCI. TNFR2 co-localized to GFAP^+^ astrocytes and to CD68^+^ and CD11b^+^ microglia/macrophages located in the peri-lesion site at 21 and 28 days after SCI in mice. This was supported by our findings of TNFR2 expression by GFAP^+^ astrocytes and Iba1^+^ microglia/macrophages in the human spinal cord after SCI. The increase in TNFR2 expression by microglia is believed to be neuroprotective. In support of this, TNFR2 activation on microglia promotes the induction of anti-inflammatory pathways, and TNFR2 ablation in microglia led to the early onset of experimental autoimmune encephalomyelitis, an animal model of multiple sclerosis, with increased leukocyte infiltration, T cell activation, and demyelination [53,54]. Moreover, TNFR2 expression by astrocytes might be important for remyelination after SCI [55,56].

IL-1β, IL-6, and CXCL1 levels also increased rapidly in response to SCI. IL-1β and CXCL10 remained elevated throughout the study, whereas IL-6 returned to baseline levels 7 days after SCI. IL-10 increased only transiently 6 h after SCI. IL-10 is an anti-inflammatory cytokine [57,58] and sustained microglial activation can be inhibited by IL-10 [59]. Thus, the low IL-10 levels in the delayed phase of SCI may help sustain chronic inflammation following SCI [60], a hypothesis that is supported by findings of reduced inflammation, limited neuronal damage, and improved functional recovery following systemic administration of IL-10 in rats subjected to SCI [58,61]. IL-1β levels increased transiently from 6–12 h after SCI and again from day 7 until the end of the experiment, and IL-1β was also highly expressed in the human spinal cord after SCI. IL-1β is known to play a detrimental role in the progression of CNS injury and to contribute to maintaining microglial/macrophage activation [62,63,64]. IL-1knockout mice show improved locomotor activity, reduced lesion volume, and increased cell survival after SCI [64,65], and acute IL-1 receptor antagonist treatment has been shown to suppress peripheral and central inflammatory responses in SCI [66], suggesting that also IL-1 plays an important role in SCI. We suggest that the early the upregulation of IL-6 that is detected after SCI is detrimental and promotes a pro-inflammatory environment, whereas its expression in the delayed phase might be important for healing after SCI. This is based on observations showing that transiently blocking IL-6 activity in the acute phase improves functional recovery and dampens the neuroinflammatory response after SCI [67]. However, continuously blocking IL-6 signaling suppresses axonal regeneration and causes failed gliosis [68,69,70]. Our findings of a significant elevation in CXCL1 levels from 6 h after SCI are consistent with its well-known function as a chemokine that attracts peripheral immune cells to the injured cord [71] and identifies CXCL1 as an important player in the acute and chronic inflammatory response after SCI.

In the normal CNS, microglia display high expression of homeostatic markers, such as the purinergic receptor P2RY12 and the fractalkine receptor CX3CR1 [72]. During the first two weeks after SCI, microglia have been shown to proliferate extensively, and they accumulate around the lesion, where they position themselves at the interface between infiltrating leukocytes and astrocytes [73]. The first week after SCI, the lesion area has been shown to contain four different microglial subtypes; homeostatic microglia (predominant in the uninjured spinal cord) and inflammatory, dividing, and migrating nonhomeostatic microglia [74]. Inflammatory microglia, the predominant microglial cell type, were mostly associated with cell death and cytokine production, and were identified by low expression of *P2ry12*. Dividing microglia, mostly related to cell cycle, and migrating microglia also expressed low levels of *P2ry12*. In our study, we found that *P2ry12* levels decreased in the acute phase but restored to uninjured levels and even to increased levels 3 days after SCI. This change was paralleled by similar changes in *Cx3cr1*. *Itgam* levels increased from day 3. We also observed increased microglial numbers and increased activation within the lesion area between 3 and 24 h after SCI, with increased MFI for CD11b and CD45, along with increased levels of pro-inflammatory cytokines in the acute phase after SCI. This supports the findings of inflammatory and dividing nonhomeostatic microglia in the lesioned cord in the acute phase after SCI. Despite the presence of high numbers of cytokine-producing inflammatory and proliferating microglia in the acute phase after SCI, these are essential components of the neuroprotective scar that forms after SCI, and microglial depletion disrupts the glial scar formation, enhances immune infiltration, reduces neuronal and oligodendrocyte survival, and impairs locomotor recovery [73].

In the delayed phase after SCI, we observed that Gal3^+^ and CD68^+^ microglia were abundant in the peri-lesion area. CD68 expression is related to phagocytosis, and microglia-dependent pro-myelinating effects have been attributed to Gal3 expression [75], favoring a pro-regenerative phenotype that fosters myelin debris phagocytosis through TREM2 activity [76,77]. As *Trem2* expression increase only transiently 7 days after SCI in the present study, it is possible that microglia in the delayed phase after SCI remain pro-inflammatory and display reduced pro-regenerative capacity, required for proper remyelination. It is likely that the presence of increased solTNF is a key factor keeping microglia, and possibly also macrophages, in a pro-inflammatory state in the more delayed phase after SCI [20,43], i.e., from 7–10 days after injury, when myelin phagocytosing and remyelination processes are occurring [78]. This assumption is supported by previous findings in our laboratory of improved functional recovery and reduced tissue damage following topical inhibition of solTNF [28] and by recent findings using big-data integration and large-scale analytics identifying solTNF as a therapeutic target for SCI [22].

Arg1 expression after SCI (1–7 days post-SCI) has been shown to be highly specific to monocytes/macrophages (versus microglia) but displays macrophage subtype specificity [74]. As the major peripheral myeloid composition at the injury site shifts from monocytes to chemotaxis-inducing macrophages and then to inflammatory macrophages over time, there is a progressive decrease in the classic anti-inflammatory enzyme arginase 1 that is associated with increased pro-inflammatory biological processes in inflammatory macrophages. We observed a significant increase in *Arg1* expression from 24 h to 7 days after SCI, supporting findings that the lesion site becomes populated by inflammatory macrophages in the delayed phases after SCI [79]. This is also consistent with the second increase in IL-1β and TNF in the delayed phase after SCI. Thus, a future challenge is to limit the pro-inflammatory macrophage state without interfering with the beneficial effects of macrophages.

Transected axons are unable to regenerate after SCI and the glial scar, consisting of astrocytes and microglia/macrophages, is thought to be responsible for this. Therefore, manipulating the inflammatory response after SCI may help regulate the formation of the glial scar, allowing for better axonal regrow. Besides therapeutically manipulating the inflammatory response after SCI using, i.e., anti-TNF therapeutics (reviewed by [17]), transplantation with mesenchymal stem cells can inhibit excessive glial scar formation and the inflammatory response leading to improved functional outcome (reviewed by [80]). Another promising cell therapeutic approach is the use of autologous transplantation of patient-derived induced pluripotent stem cell-derived oligodendrocyte precursor cell-enriched neural stem/progenitor cells, as preclinical SCI studies have demonstrated the positive effect of these cells on robust remyelination of demyelinated axons, resulting in improved functional recovery (reviewed by [81,82]).

A limitation of the present study is that only female mice were used. The acute inflammatory profile has been demonstrated to differ between female and male mice [83,84]. Additionally, the inhibition of TNF-TNFR1 signaling has been demonstrated to be therapeutic for neuropathic pain in males but not in females [85], highlighting the importance of incorporating both male and female groups in future SCI research to account for sexual dimorphisms.

Another limitation is that only young mice were used. A recent study by Stewart et al. [83] supported the inflammation after SCI is sex-dependent both at the level of cellular recruitment and phenotype, effects of aging, however, while present, were overall less pronounced. Interestingly though, *Tnf* expression was one of the genes that differed between 4-month-old and 14-month-old SCI mice, whereas sex did not appear to affect *Tnf* expression [83]. How the temporal and cellular expression of TNF changes with age and sex after SCI remains to be elucidated.

## 5. Conclusions

Our study supports the consensus that neuroprotective immunotherapies aimed against the detrimental immune response, such as the signaling through TNFR1, might effectively suppress the chronic inflammation after SCI and improve recovery.

## Figures and Tables

**Figure 1 biology-11-00939-f001:**
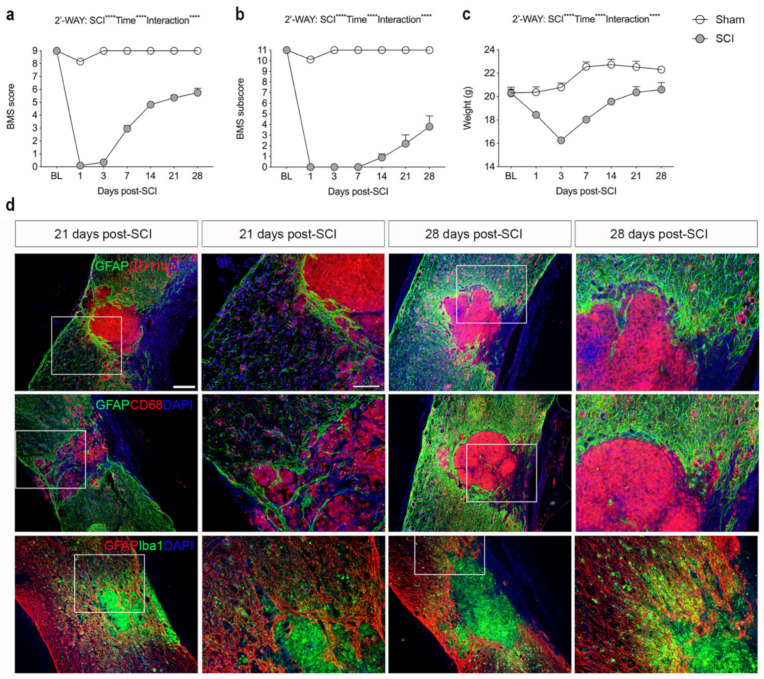
Analysis of locomotor function and glial reactions after SCI. (**a**) Evaluation of hind limb locomotor function. SCI and sham mice were tested 1, 3, and 7 days after surgery and weekly thereafter for 28 days. Motor behavior was scored under blinded conditions with the Basso Mouse Scale (BMS). Analysis of sham and SCI mice showed significantly lower BMS scores in SCI mice compared to sham mice (SCI: *p* < 0.0001, F_1,27_ = 1447; Time: *p* < 0.0001, F_6,134_ = 253.8; Interaction: *p* < 0.0001, F_6,134_ = 205.4). (**b**) Analysis of BMS subscore in SCI and sham mice demonstrating significantly lower BMS subscore in SCI mice compared to sham mice (SCI: *p* < 0.0001, F_1,27_ = 1081; Time: *p* < 0.0001, F_6,134_ = 98.31; Interaction: *p* < 0.0001, F_6,134_ = 89.62). (**c**) Body weight over time in SCI and sham mice (SCI: *p* < 0.0001, F_1,27_ = 25.71; Time: *p* < 0.0001, F_3_._36_ = 91.98; Interaction: *p* < 0.0001, F_6,134_ = 53.4). Results are expressed as mean ± SEM, *n*-values; SCI, *n* = 39 for baseline (BL) to 3 days, *n* = 29 for 7 days, *n* = 20 for 14 and 21 days, and *n* = 10 for 28 days. Sham, *n* = 20 for baseline (BL) to 3 days, *n* = 15 for 7 days, *n* = 10 for 14 and 21 days, and *n* = 5 for 28 days. (**d**) Sections of the thoracic spinal cords were double-labeled for GFAP (green; upper and middle panels, red; lower panels) and CD11b (red; upper panel), CD68 (red; middle panel), or Iba1 (green; lower panel). DAPI was used as a nuclear marker. Scale bars: low magnification = 100 μm and high magnification = 40 μm. GFAP, glial fibrillary acidic protein; CD, cluster of differentiation; Iba1, ionized calcium binding adaptor molecule 1.

**Figure 2 biology-11-00939-f002:**
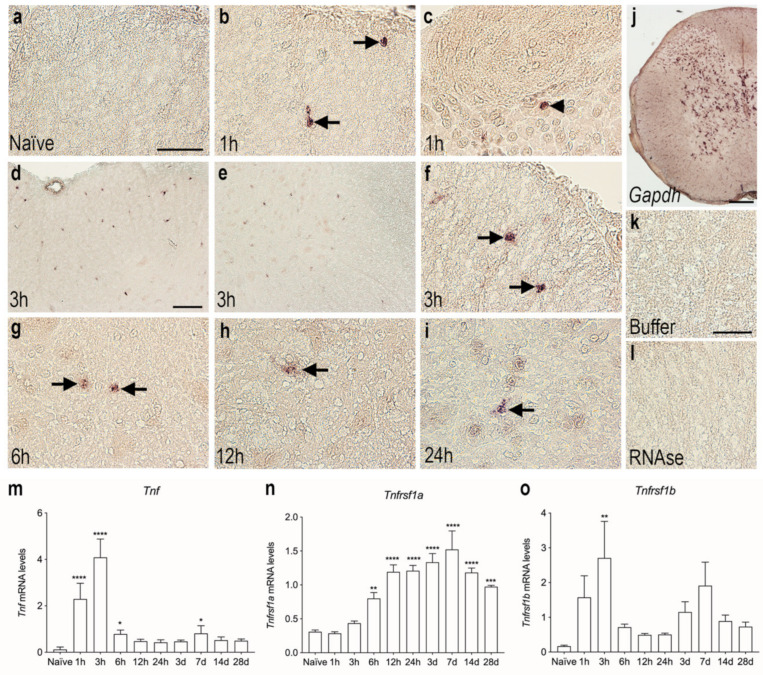
Cellular and temporal expression of *Tnf* mRNA in the thoracic spinal cord after SCI. (**a**–**i**) In situ hybridization was used to investigate the distribution of *Tnf* mRNA^+^ cells the first 24 h after SCI. *Tnf* mRNA expression was undetectable in naïve mice (**a**). *Tnf* mRNA^+^ cells in the white matter of the posterior funiculi (arrows in **b**) and in neuronal-like cells in the dorsal horn (arrowhead in **c**), at 1 h after SCI. *Tnf* mRNA^+^ cells in the white matter of the posterior funiculi (**d**) and in the grey matter of the ventral horn (**e**). At 3 h, most cells displayed macrophage- or glial-like morphology (arrows in **f**). By 6 h (**g**), 12 h (**h**), and 24 h (**i**), *Tnf* mRNA^+^ cells were mainly located in white matter areas of the damaged spinal cord (arrows). (**j**) Parallel spinal cord sections that were in situ hybridized for glyceraldehyde-3-phosphate dehydrogenase (*Gapdh*) mRNA showed a largely neuronal signal and confirmed the overall suitability of the tissue for in situ hybridization. (**k**,**l**) Parallel sections hybridized with buffer alone (**k**) or pretreated with RNAse A before the in situ hybridization (**l**) were devoid of signal. (**m**–**o**) RT-qPCR analysis of *Tnf* mRNA (**m**)*, Tnfrsf1a* mRNA (**n**), and *Tnfrsf1b* mRNA (**o**) levels in naïve mice and in mice allowed 1, 3, 6, and 12 h and 1, 3, 7, 14, and 28-days survival after SCI. *Tnf* mRNA levels were significantly increased at 1, 3, and 6 h and 7 days after SCI, compared to naïve mice (Time: *p* < 0.0001, F_9,39_ = 58.14) (**m**). *Tnfrsf1a* mRNA levels significantly increased from 6 h to 28 days after SCI, compared to naïve mice (Time: *p* < 0.0001, F_9,39_ = 20.01) (**n**). *Tnfrsf1b* mRNA levels significantly increased at 3 h after SCI, compared to naïve mice (Time: *p* = 0.009, F_9,39_ = 2.965). Results are expressed as mean ± SEM, *n* = 5/group, * *p* < 0.05, ** *p* < 0.01, *** *p* < 0.001, **** *p* < 0.0001. Scale bars: (**a**–**c**, **f**–**i**, **k**,**l**) = 40 μm, (**d**,**e**) = 100 μm, and (**j**) = 200 μm.

**Figure 3 biology-11-00939-f003:**
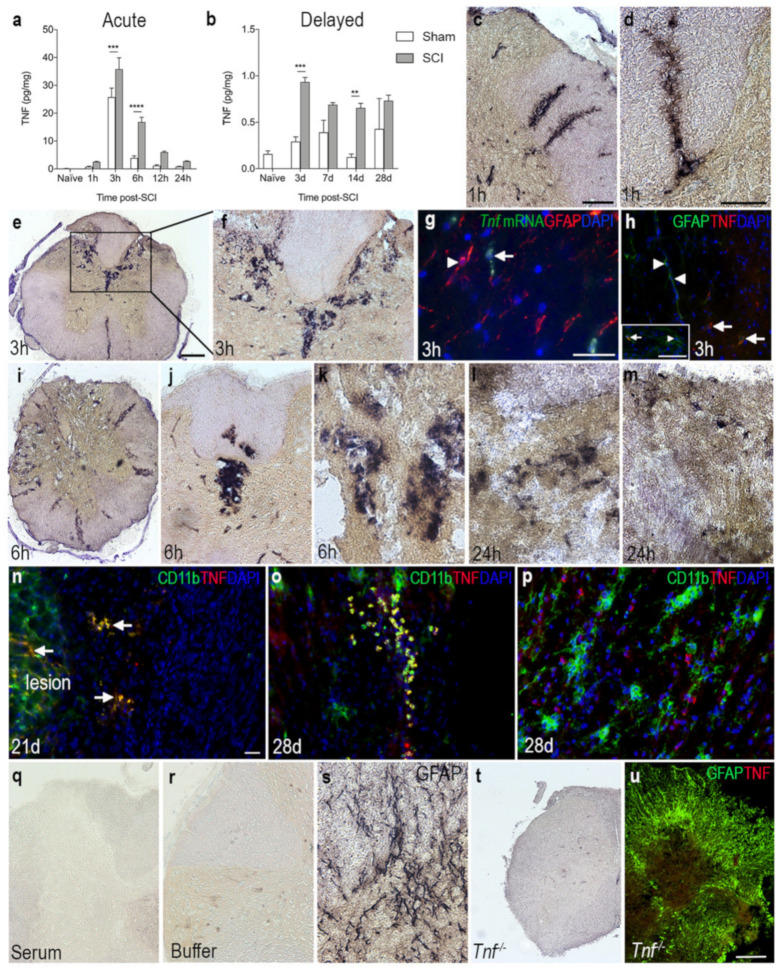
Overview of the spatiotemporal expression of TNF after SCI. (**a**,**b**) Temporal expression of TNF levels in the acute (**a**) and delayed (**b**) phase after SCI. Statistical indications represent comparisons to sham mice in the acute phase (SCI: *p* < 0.0001, F_1,48_ = 31.7; Time: *p* < 0.0001, F_5,48_ = 105.4; Interaction: *p* = 0.0008, F_5,48_ = 5.08) and delayed phase (Interaction: F_4,39_ = 3.05, *p* = 0.03; Time: F_4,39_ = 5.22, *p* = 0.002; SCI: F_1,39_ = 30.74, *p* < 0.0001) after SCI. Results are expressed as mean ± SEM, *n* = 5/group, ** *p* < 0.01, *** *p* < 0.001, **** *p* < 0.0001. (**c**,**d**) Representative images of TNF immunoreactivity in grey matter of the dorsal horn and the lateral funiculus (**c**), as well as the posterior funiculus (d) of mice that survived 1 h after SCI. (**e**,**f**) At 3 h, TNF immunoreactivity was high in the dorsal parts of the lesioned spinal cord. (**g**) Combined in situ hybridization *for Tnf* mRNA (green, arrow) and immunohistochemistry for astroglial GFAP (red, arrowhead) 3 h after SCI. (**h**) Double immunofluorescent staining for astroglial GFAP (green, arrow heads) and TNF (red, arrows) 3 h after SCI. Insert demonstrates that a minority of GFAP^+^ cells (green, arrowhead) co-expressed TNF (red, arrow). (**i**–**m**) Representative images of TNF immunoreactivity 6 h (**i**–**k**) and 24 h (**l**,**m**) after SCI. (**n**) Double immunofluorescent staining for TNF (red, arrow) and the microglial/macrophage marker CD11b (green) showed colocalization of TNF on CD11b^+^ cells near the lesion 21 days after SCI. (**o**) Double immunofluorescent staining for TNF (red) and the CD11b (green) showed colocalization of TNF on CD11b^+^ cells located at the peri-lesion area 28 days after SCI. DAPI (blue) was used as a nuclear marker. (**p**) TNF was also found in CD11b^−^ cells 28 days after SCI. (**q**,**r**) Control reactions demonstrating absence of specific staining in tissue sections incubated with rabbit IgG (**q**) or with buffer alone (**r**). (**s**) Spinal cord section stained for astrocytic GFAP as a control for the immunohistochemical reaction. (**t**,**u**) Thoracic spinal cord tissue sections from a *Tnf^−/−^* mouse 3 h after SCI, demonstrating absence of specific TNF staining. Scale bars: (**c**,**f**,**j**,**q)** = 40μm, (**e**,**i**,**t**,**u**,**o**) = 100μm, and (**d**,**g**,**h**,**k**,**l**–**n**,**p**,**r**,**s**) = 200μm. CD, cluster of differentiation; GFAP, glial fibrillary acidic protein; TNF, tumor necrosis factor.

**Figure 4 biology-11-00939-f004:**
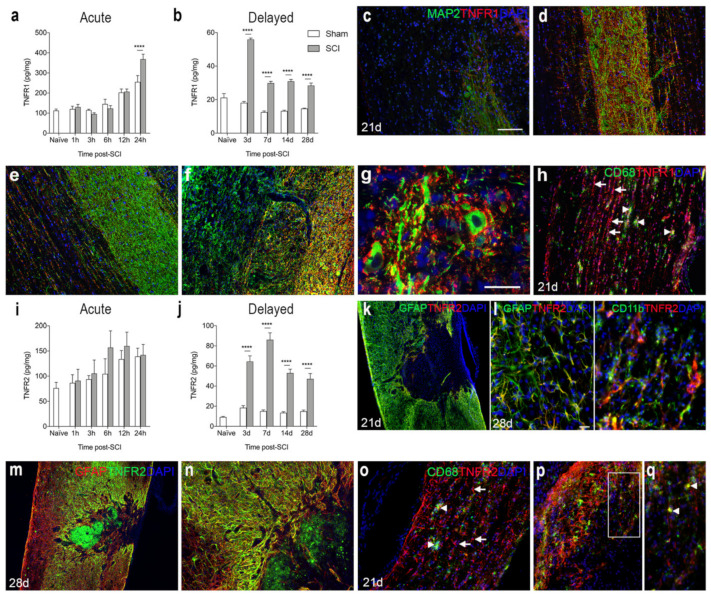
Spatiotemporal expression of TNF receptor 1 and 2 after SCI. (**a**,**b**) TNFR1 protein levels in the acute (**a**) and delayed (**b**) phases after SCI. Significant indications represent comparisons to sham mice in the acute phase (**a**, Interaction: F_5,48_ = 4.52, *p* = 0.002; Time: F_5,48_ = 45.74, *p* < 0.0001; SCI: F_1,48_ = 2.19, *p* = 0.15) and delayed phase (**b**, Interaction: F_4,40_ = 49.51, *p* < 0.0001; Time: F_4,40_ = 52.17, *p* < 0.0001; SCI: F_1,40_ = 406.5, *p* < 0.0001) after SCI. (**c**–**g**) Immunofluorescent double labelling for TNFR1 (red) and neuronal MAP2 (green) within the lesion site (**c**), peri-lesion area (**d**,**f**,**g**), as well as distant from the lesion site (**e**) at 21 (**c**–**e**) and 28 (**f**,**g**) days after SCI. (**h**) Immunofluorescent double labelling for TNFR1^+^ (red) and CD68^+^ (green) cells 21 days after SCI. (**i**,**j**) TNFR2 protein levels in the acute phase (**i**, Interaction: F_5,47_ = 0.42, *p* = 0.83; Time: F_5,47_ = 3.89, *p* < 0.005; SCI: F_1,47_ = 1.75, *p* = 0.19) and delayed phase (**j**, Interaction: F_4,40_ = 25.04, *p* < 0.0001; Time: F_4,40_ = 36.07, *p* < 0.0001; SCI: F_1,40_ = 2702.8, *p* < 0.0001) after SCI. (**k**–**n**) Immunofluorescent double labelling for TNFR2 (red: **k**,**l** and green: (**m**,**n**) and GFAP^+^ astrocytes (green: (**k**,**l**) and red: (**m**,**n**)) and CD11b^+^ microglia/macrophages (green: (**l**)) at 21 (**k**) and 28 (**l**–**n**) days after SCI. (**o**–**q**) Immunofluorescent double labeling of TNFR2^+^ cells (arrows in (**o**)) and CD68^+^ microglia/macrophages (arrow heads in (**o**)). (**p**) Only a few CD68^+^ cells co-expressed TNFR2. (**q**) represents a high magnification image of the area squared in (**p**), demonstrating TNFR2 ^+^ CD68^+^ cells located in the peri-lesion areas 21 days after SCI (arrow heads in (**q**)). DAPI was used as a nuclear marker. Scale bars: (**k**,**m**) = 100 µm, (**c**–**f**,**h**,**n**–**p**) = 40 μm, and (**g**,**l**,**q**) = 20 μm. Results are expressed as mean ± SEM, *n* = 5/group, **** *p* < 0.0001. GFAP, glial fibrillary acidic protein; MAP2, microtubule associated protein 2; TNFR, tumor necrosis factor receptor.

**Figure 5 biology-11-00939-f005:**
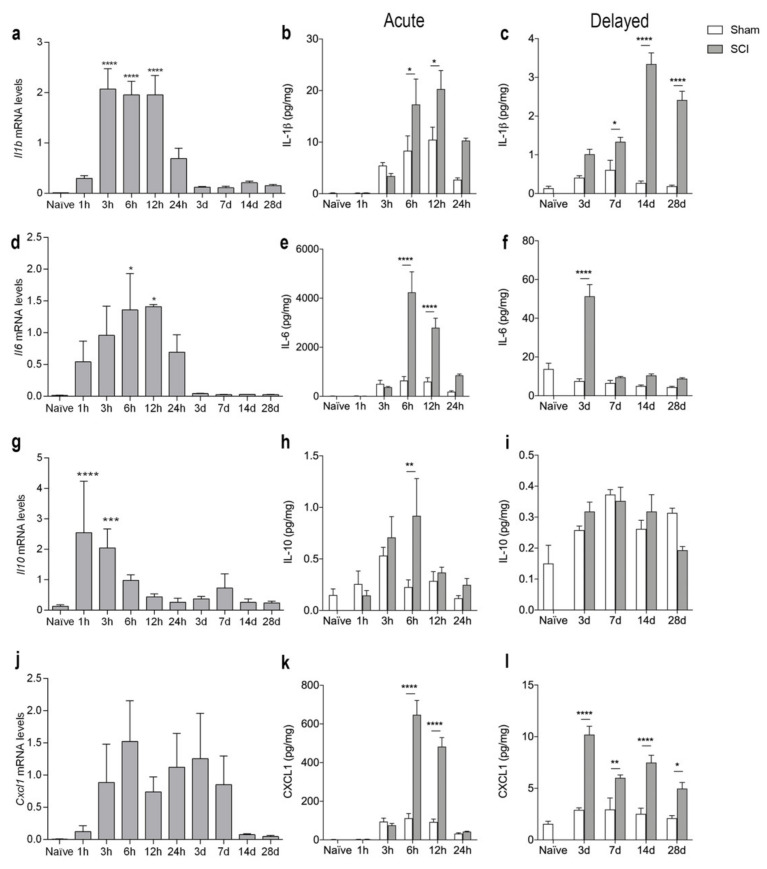
Temporal expression of inflammatory cytokines after SCI. (**a**–**c**) Temporal expression of *Il1b* mRNA ((**a**), *p* < 0.0001, Time: F_9,39_ = 17.04) and IL-1β levels in the acute phase ((**b**), Interaction: F_5,47_ = 3.07, *p* = 0.02; Time: F_5,47_ = 17.76, *p* < 0.0001; SCI: F_1,47_ = 10.86, *p* = 0.002) and delayed phase ((**c**), Interaction: F_4,39_ = 32.72, *p* < 0.0001; Time: F_4,39_ = 32.06, *p* < 0.0001; SCI: F_1,39_ = 175.6, *p* < 0.0001) after SCI. (**d**–**f**) Temporal expression of *Il6* mRNA ((**d**), Time: F_9,34_ = 3.33, *p* = 0.005) and IL-6 levels in the acute phase ((**e**), Interaction: F_5,48_ = 14.96, *p* < 0.0001; Time: F_5,48_ = 25.49, *p* < 0.0001; SCI: F_1,48_ = 42.99, *p* < 0.0001) and delayed phase ((**f**), Interaction: F_4,39_ = 34.84, *p* < 0.0001; Time: F_4,39_ = 39.20, *p* < 0.0001; SCI: F_1,39_ = 65.39, *p* < 0.0001) after SCI. (**g**–**i**) Temporal expression of *Il10* mRNA ((**g**), Time: F_9,39_ = 9.648, *p* < 0.0001) and IL-10 levels in the acute phase ((**h**), Interaction: F_5,48_ = 2.10, *p* = 0.08; Time: F_5,48_ = 4.58, *p* < 0.002; SCI: F_1,48_ = 4.21, *p* < 0.05) and delayed phase ((**i**), Interaction: F_4,39_ = 2.56, *p* = 0.05; Time: F_4,39_ = 13.23, *p* < 0.0001; SCI: F_1,39_ = 0.06, *p* = 0.81) after SCI. (**j**–**l**) Temporal expression of *Cxcl1* mRNA ((**j**), Time: F_9,39_ = 1.786, *p* = 0.1023) and CXCL1 levels in the acute phase ((**k**), Interaction: F_5,48_ = 39.83, *p* < 0.0001; Time: F_5,48_ = 69.33, *p* < 0.0001; SCI: F_1,48_ = 93.31, *p* < 0.0001) and delayed phase ((**l**), Interaction: F_4,39_ = 10.89, *p* < 0.0001; Time: F_4,39_ = 22.51, *p* < 0.0001; SCI: F_1,39_ = 96.25, *p* < 0.0001) after SCI. Results are expressed as mean ± SEM, *n* = 5/group, * *p* < 0.05, ** *p* < 0.01, *** *p* < 0.001, **** *p* < 0.0001. One technical outlier was excluded in the day 28 sham group. For mRNA analysis, significant indications represent comparisons to naïve conditions and for protein analyses comparisons to sham mice.

**Figure 6 biology-11-00939-f006:**
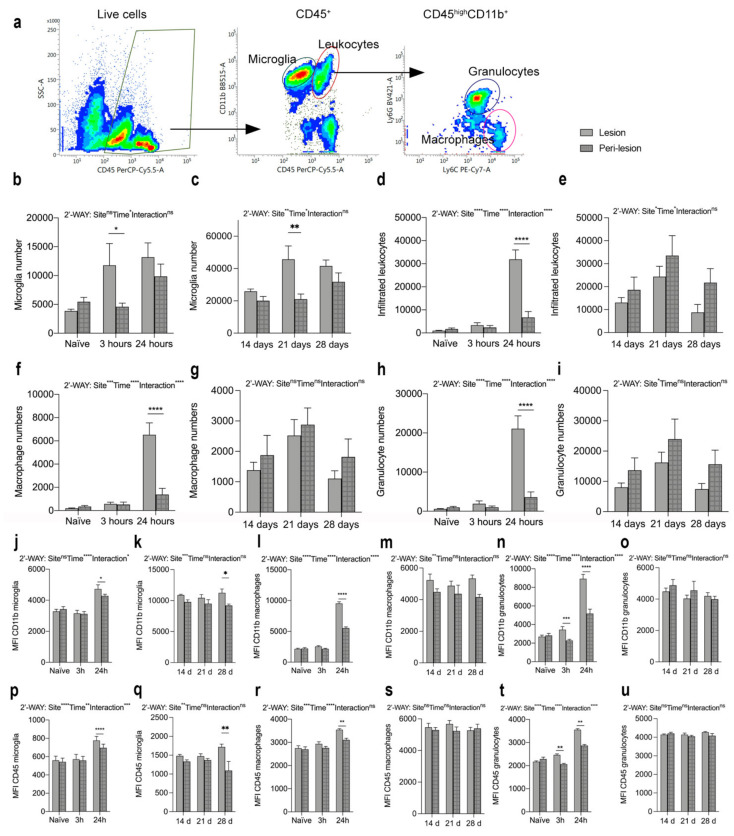
Changes in microglia and leukocyte populations after SCI. (**a**) Representative dot plots showing the gating strategy. Only live cells were included. (**b**,**c**) Number of microglia (CD11b^+^CD45^dim^) in the acute ((**b**), Interaction: F_2,28_=35.16, *p* < 0.0001; Time: F_2,28_ = 18.19, *p* < 0.0001; Site: F_1,28_ = 10.26, *p* = 0.003) and chronic ((**c**), Interaction: F_2,22_ = 2.04, *p* = 0.15; Time: F_2,22_ = 4.37, *p* = 0.03; Site: F_1,22_ = 11.14, *p* = 0.003) phases after SCI. (**d**,**e**) Number of infiltrating leukocytes (CD11b^+^CD45^high^) in the acute ((**d**), Interaction: F_2,28_ = 80.03, *p* < 0.0001; Time: F_2,28_ = 154.9, *p* < 0.0001; Site: F_1,28_ = 32.13, *p* < 0.0001) and chronic ((**e**), Interaction: F_2,22_ = 0.24, *p* = 0.79; Time: F_2,22_ = 4.09, *p* = 0.03; Site: F_1,22_ = 4.44, *p* < 0.05) phases after SCI. (**f**,**g**) Number of infiltrating Ly6C^+^Ly6G^−^ macrophages in the acute ((**f**), Interaction: F_2,28_ = 31.60, *p* < 0.0001; Time: F_2,28_ = 66.72, *p* < 0.0001; Site: F_1,28_ = 17.75, *p* = 0.0002) and chronic ((**g**), Interaction: F_2,22_ = 0.06, *p* = 0.94; Time: F_2,22_ = 3.33, *p* = 0.05; Site: F_1,22_ = 1.54, *p* = 0.23) phases after SCI. (**h**,**i**) Number of infiltrating Ly6C^+^Ly6G^+^ granulocytes in the acute ((**h**), Interaction: F_2,28_ = 39.45, *p* < 0.0001; Time: F_2,28_ = 37.67, *p* < 0.0001; Site: F_1,28_ = 20.19, *p* = 0.0001) and chronic ((**i**), Interaction: F_2,22_ = 1.52, *p* = 0.24; Time: F_2,22_ = 0.73, *p* = 0.49; Site: F_1,22_ = 1.29, *p* = 0.27) phases after SCI. (**j**,**k**) MFI for CD11b on microglia in the acute ((**j**), Interaction: F_2,28_ = 4.19, *p* = 0.03; Time: F_2.28_ = 20.58, *p* < 0.0001; Site: F_1,28_ = 1.66, *p* = 0.21) and chronic ((**k**), Interaction: F_2,22_ = 0.92, *p* = 0.92; Time: F_2,22_ = 0.37, *p* = 0.69; Site: F_1,22_ = 14.62, *p* = 0.0009) phases after SCI. (**l**,**m**) MFI for CD11b on macrophages in the acute ((**l**), Interaction: F_2,28_ = 118.8, *p* < 0.0001; Time: F_2,28_ = 480.0, *p* < 0.0001; Site: F_1,28_ = 145.80, *p* < 0.0001) and chronic ((**m**), Interaction: F_2,22_ = 0.54, *p* = 0.59; Time: F_2,22_ = 0.27, *p* = 0.77; Site: F_1,22_ = 9,71, *p* = 0.005) phases after SCI. (**n**,**o**) MFI for CD11b on granulocytes in the acute ((**n**), Interaction: F_2,28_ = 51.38, *p* < =0.0001; Time: F_2,28_ = 62.49, *p* < 0.0001; Site: F_1,28_ = 97.29, *p* < 0.0001) and chronic ((**o**), Interaction: F_2,22_ = 0.76, *p* = 0.48; Time: F_2,22_ = 1.92, *p* = 0.17; Site: F_1,22_ = 0.88, *p* = 0.36) phases after SCI. (**p**,**q**) MFI for CD45 on microglia in the acute ((**p**), Interaction: F_2,28_ = 10.41, *p* = 0.0004; Time: F_2,28_ = 5.50, *p* < 0.01; Site: F_1,28_ = 27.81, *p* < 0.0001) and chronic ((**q**), Interaction: F_2,22_ = 3.03, *p* = 0.07; Time: F_2,22_ = 0.01, *p* = 0.99; Site: F_1,22_ = 9.55, *p* = 0.005) phases after SCI. (**r**,**s**) MFI for CD45 on macrophages in the acute ((**r**), Interaction: F_2,28_ = 3.01, *p* = 0.07; Time: F_2,28_ = 29.90, *p* < 0.0001; Site: F_1,28_ = 11.08, *p* = 0.003) and chronic ((**s**), Interaction: F_2,22_ = 0.73, *p* = 0.50; Time: F_2,22_ = 0.14, *p* = 0.87; Site: F_1,22_ = 0.76, *p* = 0.39) phases after SCI. (**t**,**u**) MFI for CD45 on granulocytes in the acute ((**t**), Interaction: F_2,28_ = 33.17, *p* < =0.0001; Time: F_2,28_ = 255.9, *p* < 0.0001; Site: F_1,28_ = 63.45, *p* < 0.0001) and chronic ((**u**), Interaction: F_2,22_ = 1.52, *p* = 0.24; Time: F_2,22_ = 0.73, *p* = 0.49; Site: F_1,22_ = 1.29, *p* = 0.27) phases after after SCI. Open bars represent the lesion site, checkered bars represent the peri-lesion site. Results are expressed as mean ± SEM, *n* = 5–11/group, * *p* < 0.05, ** *p* < 0.01, *** *p* < 0.001, and **** *p* < 0.0001.

**Figure 7 biology-11-00939-f007:**
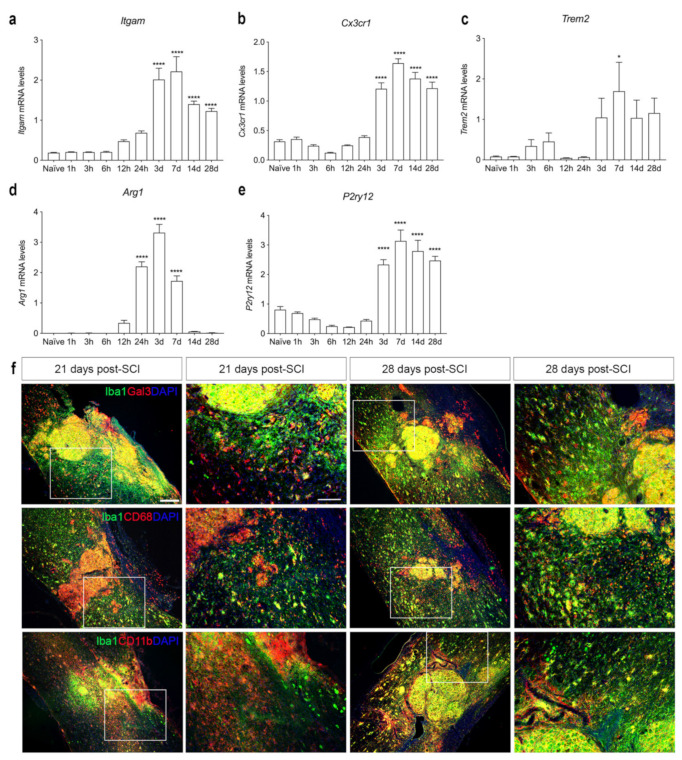
Characterization of microglia/macrophage reactions after SCI. (**a**–**e**) Temporal expression of *Itgam* ((**a**), Time: F_9,38_ = 30.97, *p* < 0.0001), *Cx3cr1* ((**b**), Time: F_9,38_ = 77.32, *p* < 0.0001), *Trem2* ((**c**), Time: F_9,37_ = 3.218, *p* = 0.006), *Arg1* ((**d**), Time: F_9,39_ = 113.6, *p* < 0.001), and *P2ry12* ((**e**), Time: F_9,38_ = 48.17, *p* < 0.001) mRNA levels after SCI. Results are presented as mean ± SEM, *n* = 5/group, * *p* < 0.05 and **** *p* < 0.0001. (**f**) Immunofluorescence double labeling of Iba1^+^ (green) and Gal3^+^ (red, upper panel), CD68^+^ (red, middle panel), or CD11b^+^ (red, lower panel) cells at 21 and 28 days after SCI. High magnification images represent squared areas in the low magnification images. Scale bars: low magnification = 100 μm and high magnification = 40 μm. CD, cluster of differentiation; Gal3, galectin-3; Iba1, ionized calcium binding adaptor molecule 1.

**Figure 8 biology-11-00939-f008:**
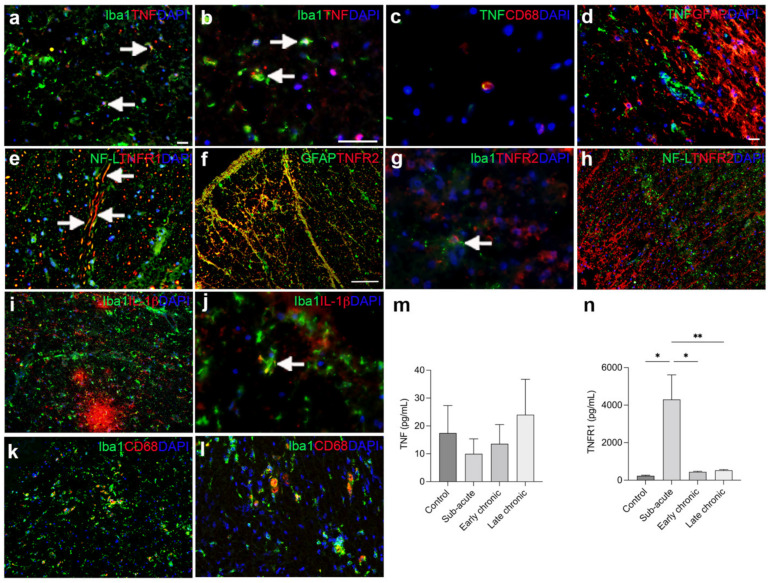
Characterization of TNF and TNFR in individuals with SCI. (**a**,**b**) Immunofluorescent double labeling of TNF (red) and Iba1 (green) expressing microglia/macrophages (arrows) in the spinal cord of an 33-year-old man with a 3-week-old C6–7 SCI. (**c**) Immunofluorescent double labeling of TNF (green) and CD68 (red) phagocytic cells in the spinal cord of an 65-year-old man with a 5-week-old C5 SCI. (**d**) Immunofluorescent double labeling of TNF^+^ (green) cells and GFAP^+^ astrocytes (red) in a 61-year-old man with a 2-week-old C1-2 SCI. (**e**) Immunofluorescent double labeling of TNFR1^+^ (red) and NF-L^+^ (green) neuronal fibers in an 67-year-old man with a 6-week-old C5-7 SCI. (**f**) Immunofluorescent double labeling of TNFR2 (red) and astroglial GFAP (green) in an 80-year-old woman with a 2-month-old C8-T1 SCI. (**g**) Immunofluorescent double labelling of TNFR2^+^ (red) and Iba1^+^ (green) microglia (arrow) in a 67-year-old man with a 6-week-old C5-7 SCI. (**h**) Immunofluorescent double labeling of TNFR2^+^ (red) cells and NF-L^+^ (green) neuronal fibers in an 33-year-old man with a 3-week-old C6-7 SCI. (**i**,**j**) Immunofluorescent double labeling of IL-1β (red) and Iba1 (green) in a 65-year-old man with a 5-week-old C5 SCI. IL-1β was found to co-localize to a subpopulation of Iba1^+^ microglia (arrow in h). (**k**,**l**) Immunofluorescent double labeling of CD68 (red) and Iba1 (green) in a 67-year-old man with a 6-week-old C5-7 SCI (k) and a 33-year-old man with a 3-week-old C6-7 SCI (l). Scale bars: (**a**,**d**,**e**,**i**,**l**) = 20 μm; (**b**,**c**,**g**,**j**) = 40 μm; (**f**,**g**,**h**) = 100 μm. (**m**,**n**) CSF TNF (**m**) and TNFR1 (**n**), Time: F_3,30_ = 10.33, *p* = 0.003) levels in individuals with SCI. Results are expressed as mean ± SEM, *n* = 5–12/group, * *p* < 0.05,** *p* < 0.01. CD, cluster of differentiation; GFAP, glial fibrillary acidic protein; Iba1, ionized calcium-binding adaptor molecule 1; IL, interleukin; NF-L, neurofilament light chain; TNF, tumor necrosis factor; TNFR, TNF receptor.

**Table 1 biology-11-00939-t001:** Gender and age distribution of SCI cases included for CSF analysis.

	Controls	Sub-Acute	Early Chronic	Late Chronic
Number of cases	5	12	10	11
Sex, n (%) men	4 (80)	11 (92)	10 (100)	11 (100)
Age, years, median (IQR)	31 (23.0; 44.5)	30.5 (28.3; 47.0)	30.0 (26.8; 36.3)	45.0 (39.0; 54.0)

**Table 2 biology-11-00939-t002:** Gender and age distribution of SCI cases included for immunofluorescent analysis.

Case	Age/Sex	Level of Injury	Cause of Injury	Post-SCI Survival Time
#1	80/F	C6–T1	Fall	15 h
#2	61/M	C1–2	Dive accident	2 weeks
#3	43/M	C7	Fall	16 days
#4	33/M	C6–7	MVA	3 weeks
#5	65/M	C4	MVA	5 weeks
#6	67/M	C5–7	Fall	6 weeks

Abbreviations: C—cervical; F—female; M—male; MVA—motor vehicle accident; T—thoracic.

**Table 3 biology-11-00939-t003:** Primers used for RT-qPCR analysis.

Gene	Primer Sequences (5′-3′)	Accession No.
*Tnf*	F- AGGCACTCCCCCAAAAGATG	NM_001278601.1
	R- TCACCCCGAAGTTCAGTAGACAGA	
*Tnfrsf1a*	F- GCCCGAAGTCTACTCCATCATTTG	NM_011609.4
	R- GGCTGGGGAGGGGGCTGGAGTTAG	
*Tnfrsf1b*	F- GCCCAGCCAAACTCCAAGCATC	NM_011610.3
	R- TCCTAACATCAGCAGACCCAGTG	
*Il1b*	F- TGCCACCTTTTGACAGTGATG	NM_008361.4
	R- CAAAGGTTTGGAAGCAGCCC	
*Il6*	F- AGGATACCACTCCCAACAGA	NM_001314054.1
	R- ACTCCAGGTAGCTATGGTACTC	
*Il10*	F- GCCAGGTGAAGACTTTCTTTCAAAC	NM_010548.2
	R- AGTCCAGCAGACTCAATACACAC	
*Cxcl1*	F- GCTGGGATTCACCTCAAGAAC	NM_008176.3
	R- TGTGGCTATGACTTCGGTTTG	
*Itgam*	F- GCCTGTCACACTGAGCAGAA	NM_008401.2
	R- TGCAACAGAGCAGTTCAGCA	
*Cx3cr1*	F- TCCCATCTGCTCAGGACCTC	NM_009987.4
	R- GGCCTCAGCAGAATCGTCAT	
*Trem2*	F- TGCTGGAGATCTCTGGGTCC	NM_031254.3
	R- AGGTCTCTTGATTCCTGGAGGT	
*Arg1*	F- ATGAAGAGCTGGCTGGTGTG	NM_007482.3
	R- CCAACTGCCAGACTGTGGTC	
*P2ry12*	F- GCCAGTGTCATTTGCTGTCAC	NM_027571.4
	R- TAGATGCCACCCCTTGCACT	
*Hprt1*	F- TCCTCAGACCGCTTTTTGCC	NM_013556.2
	R- TCATCATCGCTAATCACGACGC	

**Table 4 biology-11-00939-t004:** Overview of antibodies used for immunohistochemistry and immunofluorescent staining.

Antibody	Conjugated	Host (Clone)	Dilution	Source (cat. no.)
Anti-TNF	Unconjugated	Rabbit	1:200	Thermo Fischer Scientific (P-350)
Anti-TNFR1	Unconjugated	Rabbit (H-271)	1:50	Santa Cruz (sc-7895)
Anti-TNFR2	Unconjugated	Rabbit	1:200	Sigma Aldrich (HPA004796)
Anti-TNFR2	Unconjugated	Goat	1:50	R&D Systems (AF-426-PB)
Anti-MAP2	Unconjugated	Chicken	1:100	Abcam (ab5392)
Anti-CD68	Unconjugated	Rat	1:400	Bio-Rad (MCA1957)
Anti-CD68	Unconjugated	Mouse (PG-M1)	1:100	Abcam (ab783)
Anti-CD11b	Unconjugated	Rat	1:500	Bio-Rad (MCA711)
Anti-Iba1	Unconjugated	Rabbit	1:500	Wako (019–19741)
Anti-IBA1	Unconjugated	Mouse (GT10312)	1:1000	Sigma Aldrich (SAB2702364)
Anti-Galectin-3	Unconjugated	Rat (M38)	1:300	Hakon Leffler’s Lab [38]
Anti-NF-L	Alexa Fluor-488	Mouse (N52)	1:50	Sigma Aldrich (MAB5266)
Anti-IL-1β	Unconjugated	Mouse (2E8)	1:50	Bio-Rad (MCA5542Z)
Anti-GFAP	Cy3	Mouse (G-A-5)	1:500	Sigma Aldrich (C9205)
Anti-GFAP	Alexa Fluor-488	Mouse (131–17719)	1:400	Invitrogen (A21294)
Anti-Rabbit	Alexa Fluor-594	Donkey	1:200	Invitrogen (A21207)
Anti-Rabbit	Alexa Fluor-488	Chicken	1:200	Invitrogen (A21441)
Anti-Rat	Alexa Fluor-594	Goat	1:200	Invitrogen (A11007)
Anti-rabbit	Alexa Fluor-594	Goat	1:200	Invitrogen (A21207)
Anti-rabbit	Alexa Fluor-488	Goat	1:200	Invitrogen (A11008)
Anti-rabbit	Alexa Fluor-568	Goat	1:200	Invitrogen (A11011)
Anti-Rat	Alexa Fluor-488	Goat	1:200	Invitrogen (A11006)
Anti-Chicken	Alexa Fluor-488	Goat	1:200	Invitrogen (A11039)
Anti-mouse	Alexa Fluor-488	Goat	1:200	Invitrogen (A11001)
Anti-mouse	Alexa Fluor-568	Goat	1:200	Invitrogen (A11004)
Anti-mouse	Alexa Fluor-555	Goat	1:200	Invitrogen (A21422)
Anti-goat	Alexa Fluor-594	Donkey	1:200	Invitrogen (A11058)

**Table 5 biology-11-00939-t005:** Overview of antibodies used for flow cytometry.

Antibody	Conjugated	Host (Clone)	Dilution	Source (cat. no.)
Anti-CD45	PerCP-Cy5.5	Rat (30-F11)	1:100	BD Biosciences (561869)
Anti-CD11b	BB515	Rat (M1/70)	1:200	BD Biosciences (564454)
Anti-Ly-6C	PE-Cy7	Rat (AL-21)	1:200	BD Biosciences (560593)
Anti-Ly-6G	BV421	Rat (1A8)	1:200	BD Biosciences (562737)
IgG2b, κ	PerCP-Cy5.5	Rat (A95–1)	1:100	BD Biosciences (550764)
IgG2b, κ	BB515	Rat (A95–1)	1:200	BD biosciences (564421)
IgM, κ	PE-Cy7	Rat (clone R4–22)	1:200	BD biosciences (560572)
IgG2a, κ	BV421	Rat (clone R35–95)	1:200	BD Biosciences (562602)

## Data Availability

Requests to access datasets should be directed to klambertsen@health.sdu.dk.

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
