# Peer review of "The Inflammatory Response after Moderate Contusion Spinal Cord Injury: A Time Study"

_biology, 2022, doi:10.3390/biology11060939_

Round 1
Reviewer 1 Report
Review of manuscript by Lund et al., titled “The inflammatory response after moderate contusion spinal cord injury: A time study”.
In this manuscript authors have studied the examined activation, recruitment, and polarization of microglia and infiltrating immune cells, focusing specifically on tumor necrosis factor (TNF) and its receptors, TNFR1 and TNFR2.
Following are the concerns:
- There are lot of previous studies which showed increased TNF and glial activation after SCI. Also, study has shown role of TNFR1 deletion in recovery after traumatic spinal cord injury (PMID: 11517251). Hence novelty is an issue. Authors should clearly mention rationale and the importance of the current study in instruction as well as discussion.
- Line # 47-48, this statement is very broad “neutralizing detrimental immune signaling”. Please specify exactly what authors want to target based on their outcome/data.
- Provide details of the age/gender/SCI status of the Human SCI tissue (line#160). For the Human CSF study, CSF has been taken from the patients age range 23 to 54 years. However, Human SCI tissue study, postmortem human spinal cord samples were obtained from 33 to 80 years. Another point is that for human CSF study, all the patients were men. However, postmortem human spinal cord samples were obtained from men and women both. Can age/gender affect the results? Please discuss this in detail.
- Again, for the animal study, female C57BL6/J mice were used. Please clarify?
- Flow cytometry is performed only at acute phase of the SCI. Please provide the flow data of the late phase of SCI.
- Figure 8 is very confusing; all the groups are not represented, the missing images/groups are Iba1+ TNFR1, NF-L +TNF, NF-L+TNFR2, GFAP+TNF and GFAP +TNFR1.
Author Response
Reviewer #1:
Review of manuscript by Lund et al., titled “The inflammatory response after moderate contusion spinal cord injury: A time study”.
In this manuscript authors have studied the examined activation, recruitment, and polarization of microglia and infiltrating immune cells, focusing specifically on tumor necrosis factor (TNF) and its receptors, TNFR1 and TNFR2.
Following are the concerns:
- There are lot of previous studies which showed increased TNF and glial activation after SCI. Also, study has shown role of TNFR1 deletion in recovery after traumatic spinal cord injury (PMID: 11517251). Hence novelty is an issue. Authors should clearly mention rationale and the importance of the current study in instruction as well as discussion.
We agree with the reviewer that a lot of previous studies have showed increased TNF and glial activation after SCI. We have included a recent reference on a systematic review on TNF expression after SCI performed by our lab (PMID: 35604578). During the writing of this review, it became apparent to us that though a lot of studies have investigated TNF expression and the effect of TNF deficiency, as well as TNFR1 and TNFR2 deficiency, in SCI, no paper systematically investigated TNF, TNFR1, and TNFR2 expression alongside other inflammatory markers in the acute, subacute, and chronic phases after SCI. As requested by the reviewer, we modified the introduction to mention the rationale, so that this now reads “As TNF is believed to be one of the most promising neuroinflammatory targets in SCI [17], we, in this study, investigated the temporal and cellular source of TNF and its two receptors in the acute and delayed phases after SCI using a moderate contusive SCI model in C57BL/6J mice.” on page 3, lines 94-97.
In addition, the paper mentioned above by the reviewer (PMID:11517251) was already cited in our first submission of this manuscript and is in the present revision citation #24.
- Line # 47-48, this statement is very broad “neutralizing detrimental immune signaling”. Please specify exactly what authors want to target based on their outcome/data.
As requested by the reviewer, we broadened the statement so that no reads “Diminishing detrimental neuroinflammatory processes, such as excessive production of pro-inflammatory cytokines, is considered a possible therapeutic strategy in individuals with SCI, therefore more detailed knowledge on the temporal and cellular synthesis of inflammatory mediators is required.” This has been included on page 3, lines 91-94.
- Provide details of the age/gender/SCI status of the Human SCI tissue (line#160). For the Human CSF study, CSF has been taken from the patients age range 23 to 54 years. However, Human SCI tissue study, postmortem human spinal cord samples were obtained from 33 to 80 years. Another point is that for human CSF study, all the patients were men. However, postmortem human spinal cord samples were obtained from men and women both. Can age/gender affect the results? Please discuss this in detail.
As requested by the reviewer, we included age/gender/level of injury/cause of injury, as well as survival time after SCI for the human SCI. This has been included as a new Table 2, page 5, lines 174-175.
We appreciate the reviewer’s comments on gender-specific and potential age-related differences in neuroinflammatory responses after SCI. Postmortem SCI specimens were obtained from 1 female and 5 male and CSF samples were mostly from men (80%). We have included correlation analyses on CSF TNF levels and age as well as CSF TNFR1 levels and age. We observed no association between TNF levels and age (whole group Spearman r=0.05, p=0.84; control group Spearman r=-0.50, p>0.99; subacute SCI group Spearman r=-0.5, >0.99; early chronic Spearman r=0.1, p=0.95; and late chronic Spearman r=0.49, p=0.36). We also observed no association between TNFR1 levels and age (whole group Spearman r=0.16, p=0.37; control group Spearman r=0.2, p=0.78; subacute SCI group Spearman r=-0.02, p=0.95; early chronic Spearman r=-0.46, p=0.30; and late chronic Spearman r=0.54, p=0.09). We included only the analysis on the whole group in the revised manuscript. In addition, we included a detailed discussion on gender-specific differences in the neuroinflammatory response post-SCI, page 27, lines 885-889.
- Again, for the animal study, female C57BL6/J mice were used. Please clarify?
We agree with the reviewer that there are sex-dependent differences in the inflammatory profiles after SCI. Female mice were used, as male mice are more likely to contract urinary infections and bladders are more likely to rupture during long recovery times with two daily manual bladder voiding. We included a discussion on sex and age differences in the neuroinflammatory response after SCI, page 27, lines 885-896.
- Flow cytometry is performed only at acute phase of the SCI. Please provide the flow data of the late phase of SCI.
To address this comment, we have included new flow cytometric analysis of mice subjected to SCI and allowed 14-, 21-, or 28-days survival post-SCI (n=5/group). To compare with the acute phase after SCI, we included analysis of CD11b+CD45dim microglia, CD11b+CD45high leukocytes, including CD11b+CD45highLy6G+Ly6C+ granulocytes and CD11b+CD45highLy6G-Ly6C+ monocytes. The new flow data has been incorporated in the new Figure 6 and new Supplemental Figure 1. Even though analyses are run on the same flow cytometer, data are generated 1 year apart and we therefore have chosen to present data separately from the data obtained at the acute time points.
- Figure 8 is very confusing; all the groups are not represented, the missing images/groups are Iba1+ TNFR1, NF-L +TNF, NF-L+TNFR2, GFAP+TNF and GFAP +TNFR1.
We understand the reviewer’s wish for additional human stains. As the reviewer is probably aware, human tissue from SCI patients is a unique and rare thing to get your hands on. We have chosen combinations of antibodies that replicate our mouse findings and therefore not included all combinations. We have; however, a few sections left and could accommodate to include 2 additional stains. To accommodate this point, we have included images of two new cases stained for GFAP/TNF (new Figure 8d) and TNFR2/NF-L (new Figure 8h) in the modified new Figure 8.

Reviewer 2 Report
This manuscript studies the temporal and cellular source of TNF and its two receptors in the acute and delayed phases after spinal cord injury (SCI) using mice model. The authors verified findings in postmortem tissue and cerebrospinal fluid, derived from SCI individuals. The findings of this study will be of interest to the readers of this journal. The manuscript is well written, and the methods are clearly and satisfactorily described. There are some minor points that the authors need to address:
- If deemed adequate, the authors could also add a short paragraph indicating where (i.e., in which neuronal/glial populations) are some of the genes analyzed are expressed? This would be useful for a thoughtful explanation on whether the changes induced by these are cell-autonomous or not.
- Similarly, it would be adequate, for the non-specialist reader, that the article also highlights different potential mechanisms after SCI that improve recovery. It could be useful to add on global prevalence and incidence of traumatic SCI in human population.
- It would be as well adequate to potentially include a reflection on the other pathways, and why these may not show enough compensatory power?
- Despite they may be out of scope, could the authors offer some hint on the potential changes of SCI biology in age-related changes in SCI cases?
- On many occasions, some concepts are just stated with little explanation and/or no support literature. Please, include some additional references.
- It will be nice to discuss how different model systems (e.g., iPS cells) can be used for these studies
Author Response
Reviewer #2:
This manuscript studies the temporal and cellular source of TNF and its two receptors in the acute and delayed phases after spinal cord injury (SCI) using mice model. The authors verified findings in postmortem tissue and cerebrospinal fluid, derived from SCI individuals. The findings of this study will be of interest to the readers of this journal. The manuscript is well written, and the methods are clearly and satisfactorily described. There are some minor points that the authors need to address:
- If deemed adequate, the authors could also add a short paragraph indicating where (i.e., in which neuronal/glial populations) are some of the genes analyzed are expressed? This would be useful for a thoughtful explanation on whether the changes induced by these are cell-autonomous or not.
Although this may be a challenging thing to do, we briefly mentioned the cell populations expressing the genes analyzed in the present study. We included this in the introduction in the following sentence: “In addition, we evaluated the expression profile of selected glial-derived cytokines (IL-1b, IL-6, IL-10, and CXCL1) and examined the polarization of microglia/macrophages by investigating temporal changes in microglial/macrophage specific genes (Itgam, Cx3cr1, Trem2, Arg1, P2ry12).”, page 3, lines 99-103.
- Similarly, it would be adequate, for the non-specialist reader, that the article also highlights different potential mechanisms after SCI that improve recovery. It could be useful to add on global prevalence and incidence of traumatic SCI in human population.
As requested by the reviewer, we included the following text on global prevalence and incidence in the introduction “Spinal cord injury (SCI) is a serious neurological condition with an unknown prevalence and estimated annual incidence of 40 to 80 cases per million population, according to the World Health Organization.”, page 3, lines 57-59.
Regarding highlights of potential mechanisms after SCI that improve recovery, please see comments to #6 below.
- It would be as well adequate to potentially include a reflection on the other pathways, and why these may not show enough compensatory power?
Even though this is a valid point, our data along with the literature mentioned in the discussion support TNF as the acute driver of neuroinflammation after SCI. We therefore focused our discussion on reflecting on this pathway and believe that adding additional paragraphs on the other pathways will result in a long and unfocused discussion. The roles of IL-1b, IL-6, IL-10, and CXCL1 in SCI are already discussed on page 8, lines 800-824. We hope the reviewer agrees.
- Despite they may be out of scope, could the authors offer some hint on the potential changes of SCI biology in age-related changes in SCI cases?
We have included a paragraph on considerations for studying age as a biological variable in the neuroinflammatory response after SCI. This paragraph has been included on page 27, lines 890-896.
- On many occasions, some concepts are just stated with little explanation and/or no support literature. Please, include some additional references.
We thoroughly went through the manuscript and believe that concepts are supported by the literature.
- It will be nice to discuss how different model systems (e.g., iPS cells) can be used for these studies
Even though this is out of the scope of the present manuscript, we tried to accommodate this reviewer’s request by briefly discussing the use of iPS cells as a potential new treatment that may help improve recovery after SCI, which should also accommodate the reviewer’s wish for including potential mechanisms that may improve recovery after SCI, as mentioned in reviewer’s comment #2. Please see page 26, lines 873-884.

Round 2
Reviewer 1 Report
Authors have revised the manuscript satisfactorily. I don't have further concerns from my end.